# Non-Rectangular Robust MDPs with Normed Uncertainty Sets

**Navdeep Kumar**[*]
Technion

**Adarsh Gupta**
Finsynth.ai

**Maxence Mohamed Elfatihi**
École Polytechnique

**Giorgia Ramponi**
University of Zurich

**Kfir Levy**
Technion

**Shie Mannor**
Technion

## Abstract

Robust policy evaluation for non-rectangular uncertainty set is generally NP-hard, even in approximation. Consequently, existing approaches suffer from either exponential iteration complexity or significant accuracy gaps. Interestingly, we identify a powerful class of $L_p$-bounded uncertainty sets that avoid these complexity barriers due to their structural simplicity. We further show that this class can be decomposed into infinitely many `sa`-rectangular $L_p$-bounded sets and leverage its structural properties to derive a novel dual formulation for $L_p$ robust Markov Decision Processes (MDPs). This formulation reveals key insights into the adversary's strategy and leads to the **first polynomial-time robust policy evaluation algorithm** for $L_1$-normed non-rectangular robust MDPs.

## 1 Introduction

Robust Markov Decision Processes (MDPs) effectively handle uncertainties in environmental parameters, making them indispensable for high-stakes domains such as robotics, finance, healthcare, and autonomous driving, where failures can have catastrophic consequences [24, 12, 30, 25, 16]. They also outperform standard MDPs in terms of generalization, ensuring robust performance across diverse scenarios [37, 38, 26]. Consequently, extensive research has been conducted on robust MDPs [23, 10, 36, 17, 3, 25, 16, 14, 34, 35, 33, 7, 20, 18, 39, 31, 1, 15], primarily focusing on rectangular uncertainty sets that leverage the contractive robust Bellman operator. However, practical robust MDPs often feature non-rectangular uncertainty sets, where rectangular relaxations can result in overly suboptimal solutions [21, 8, 36]. Intuitively, non-rectangular uncertainty set could be thought of as an $n$-dimensional sphere of unit radius, and its rectangular relaxation is the smallest $n$-dimensional cube encapsulating the sphere. The ratio between the sphere and the encapsulating cube is exponential in the dimension ($O(2^{-n})$) [28]). This suggests that the rectangular relaxation of the non-rectangular uncertainty set, contains many additional environments. Moreover, most of the additional environments would lie near the corners representing big differences from the center in many coordinates – scenarios unlikely to occur in the real world, as aptly captured by the paper titled "*Lightening doesn't strike twice, robust MDPs* [22]". These improbable, highly perturbed environments can lead to a significant gap between the solutions of non-rectangular robust MDPs and their rectangular relaxations.

While non-rectangular robust MDPs capture much better interdependencies across the states, they lack the existence of contractive robust Bellman operators, which makes the problem very difficult to solve with standard dynamic programming techniques [8]. This makes non-rectangular robust

---

[*]corrosponding author email `navdeep.kumar@zohomail.in`

39th Conference on Neural Information Processing Systems (NeurIPS 2025).

MDPs a crucial yet challenging area of study, with only a limited body of work existing on the topic [36, 21, 8].

The key challenge for non-rectangular robust MDPs is robust policy evaluation . That is, given oracle access to the robust gradient (robust policy evaluation ), the robust policy gradient method can efficiently achieve an $\epsilon$-close globally optimal policy with an iteration complexity of $O(\epsilon^{-4})$ [33]. However, the robust policy evaluation for general convex non-rectangular uncertainty sets is strongly NP-hard, even for approximations [36]. Despite this hardness, [21] first proposed two methods for non-rectangular robust policy evaluation for general convex uncertainty sets: One with exponential iteration complexity in the state-action space, and another that approximates the solution but with a possible very high accuracy gap in the worst case (see Table 1).These pioneering approaches remain computationally prohibitive or overly imprecise, consistent with the NP-hardness result.

Interestingly, the NP-hardness result in [36] applies specifically to kernel uncertainty sets with certain polyhedral structures (see Appendix for details). For $L_p$-bounded uncertainty sets, [8] showed that robust policy evaluation can be done efficiently, though this is limited to reward uncertainty, a much simpler case compared to kernel uncertainties. This raises a critical open question: **Are there useful classes of kernel uncertainty sets that avoid this NP-Hardness barrier?** And we show that $L_1$ normed non-rectangular uncertainty sets are efficiently solvable in polynomial time.

We identify a specific class of non-rectangular uncertainty sets, bounded by an $L_p$- ball around a nominal kernel, and demonstrate that it effectively circumvents the NP-hardness result of [36]. Moreover, we show that robust policy evaluation for this non-rectangular $L_p$- -bounded uncertainty set is equivalent to robust policy evaluation over an infinite collection of `sa`-rectangular $L_p$ -bounded uncertainty sets. While robust policy evaluation for each `sa`-rectangular set is computationally tractable [20, 18], managing this infinite collection poses significant challenges. To overcome this, we leverage the property that the worst kernel for each `sa`-rectangular uncertainty set is a rank-one perturbation of the nominal kernel [18]. This insight enables us to express the robust policy evaluation problem (or robust return) in a novel dual form, providing a clearer understanding of the adversary's behavior. Furthermore, this dual formulation facilitates the development of a binary search method for robust policy evaluation, achieving an iteration complexity of $\tilde{O}(\log \epsilon^{-1})$ for approximating the robust return up to $\epsilon$ tolerance.

In summary, robust MDPs are critical for handling uncertainties in high-stakes domains, yet existing methods are largely confined to rectangular uncertainty sets, limiting real-world applicability. Non-rectangular uncertainty sets, though more realistic, often face NP-hard challenges in robust policy evaluation. This work identifies a promising class of non-rectangular $L_p$ -bounded kernel uncertainty sets, demonstrating that they circumvent existing NP-hardness results and enable efficient robust policy evaluation. By connecting robust evaluation for these sets to an infinite collection of `sa`-rectangular uncertainty sets and leveraging their structure, we propose a computationally efficient binary search method with logarithmic iteration complexity. This approach not only advances the understanding of non-rectangular robust MDPs but also opens the door to future investigation into broader classes of non-rectangular uncertainty sets in robust MDPs.

## 2  Preliminary

A robust Markov Decision Process (RMDP) can be described as a tuple $(\mathcal{S}, \mathcal{A}, \gamma, \mu, R, \mathcal{U})$, where $\mathcal{S}$ is the state space, $\mathcal{A}$ is the action space, $\mu$ is an initial distribution over states $\mathcal{S}$, $\gamma$ is a discount factor in $[0, 1)$, $R$ is a reward function mapping $\mathcal{S} \times \mathcal{A}$ to $\mathbb{R}$, and $\mathcal{U}$ set of transition kernel $P$ that maps $\mathcal{S} \times \mathcal{A}$ to $\Delta_{\mathcal{S}}$ [16, 25] . A policy $\pi : \mathcal{S} \to \Delta_{\mathcal{A}}$ is a decision rule that maps state space to a probability distribution over action space. Let $\Pi = (\Delta_{\mathcal{A}})^{\mathcal{S}}$ denote set of all possible policies. Further, $\pi(a|s), P(s'|s, a)$ denotes the probability of taking action $a$ in state $s$ by policy $\pi$, and the probability of transition to state $s'$ from state $s$ under action $a$ respectively. In addition, we denote $P^\pi(s'|s) = \sum_a \pi(a|s)P(s'|s, a)$ and $R^\pi(s) = \sum_a \pi(a|s)R(s, a)$ as short-hands. The return of a policy $\pi$, is defined as $J_P^\pi = \langle \mu, v_P^\pi \rangle = \langle R^\pi, d_P^\pi \rangle$ where $v_P^\pi := D^\pi R^\pi$ is value function, $d_P^\pi = \mu^\top D^\pi$ is occupation measure and $D^\pi = (I - \gamma P^\pi)^{-1}$ is occupancy matrix [27]. As a shorthand, we denote the state-action occupation measure as $d_P^\pi(s, a) = d_P^\pi(s)\pi(a|s)$ and the usage shall be clear from the context. For an uncertainty set $\mathcal{U}$, the robust return $J_{\mathcal{U}}^\pi$ for a policy $\pi$, and the optimal robust return $J_{\mathcal{U}}^*$, are defined as:

$$J_{\mathcal{U}}^\pi = \min_{P \in \mathcal{U}} J_P^\pi, \quad \text{and} \quad J_{\mathcal{U}}^* = \max_\pi J_{\mathcal{U}}^\pi,$$

Table 1: Related Work on Robust Policy Evaluation for Non-Rectangular Uncertainty Sets.

| Method | Uncertainty Set | Iteration Complexity | Accuracy | Irreducibility Assumption 1 of [21] | NP-Harness Result of [36] |
|---|---|---|---|---|---|
| [8] | **Reward** $L_p$ Normed | $O(\log \epsilon^{-1})$ | $\epsilon$ | No | No |
| Algorithm 3.1 of [21] | General Kernel Set | $O(\mathbf{2^q} \log \epsilon^{-1})$ | $\epsilon$ | No | Yes |
| Algorithm 3.2 of [21] | General Kernel Set | $O(\epsilon^{-2})$ | $\boldsymbol{\delta_d(2\epsilon + \delta_{\mathcal{P}})}$ | Yes | Yes |
| **Ours** | Kernel $L_1$ Normed | $\log(\epsilon^{-1})$ | $\epsilon$ | No | **No** |

The constants $q, \delta_d, \delta_{\mathcal{P}}$ can be as large as $O(S^2 A)$, $O(2^S)$ and $O(S\sqrt{A})$ respectively.

respectively. The objective is to determine an optimal robust policy $\pi_{\mathcal{U}}^*$ that achieves the optimal robust performance $J_{\mathcal{U}}^*$. Unfortunately, even robust policy evaluation (i.e., finding the worst-case transition kernel $P_{\mathcal{U}}^\pi \in \arg\min_{P \in \mathcal{U}} J_P^\pi$) is strongly NP-hard for general (non-rectangular) convex uncertainty sets [36]. This makes solving non-rectangular robust MDPs a highly challenging problem.

To make the problem tractable, a common approach is to use s-rectangular uncertainty sets, $\mathcal{U}^{\mathsf{s}} = \times_{s \in \mathcal{S}} \mathcal{P}_s$, where the uncertainty is modeled independently across states [36]. These sets decompose state-wise, capturing correlated uncertainties within each state while ignoring inter-dependencies across states. A further simplification is the sa-rectangular uncertainty set, $\mathcal{U}^{\mathsf{sa}}$, where uncertainties are assumed to be independent across both states and actions. Formally, $\mathcal{U}^{\mathsf{sa}} = \times_{(s,a) \in \mathcal{S} \times \mathcal{A}} \mathcal{P}_{s,a}$, where $\mathcal{P}_{s,a}$ are independent component sets for each state-action pair [16, 25, 34, 35].

A $L_p$-bounded uncertainty sets, $\mathcal{U}_p^{\mathsf{sa}}$ and $\mathcal{U}_p^{\mathsf{s}}$, which are centered around a nominal transition kernel $\hat{P}$ are defined as $\mathcal{U}_p^{\mathsf{sa}} = \{P \mid \sum_{s'} P_{sa}(s') = 1, \|P_{sa} - \hat{P}_{sa}\|_p \leq \beta_{sa}\}$, and $\mathcal{U}_p^{\mathsf{s}} = \{P \mid \sum_{s'} P_{sa}(s') = 1, \|P_s - \hat{P}_s\|_p \leq \beta_s\}$, where radius vector $\beta$ is assumed small enough to ensure all kernels within the uncertainty sets are valid [14, 7, 20, 18]. Interestingly, for $L_p$ bounded uncertainty set, adversarial (worst) kernels is a rank one perturbation of the nominal kernel that is used later in the paper [18].

**Dual Formulation.** The primal formulation of an MDP is defined as:

$$\max_{v \in \mathcal{V}} \langle \mu, v \rangle, \quad \text{with its dual:} \quad \max_{d \in \mathcal{D}} \langle d, R \rangle,$$

where $\mathcal{V} = \{v \mid v = R^\pi + \gamma P^\pi v, \pi \in \Pi\}$ represents the set of value functions. The dual formulation relies on the state-action occupancy measure $d$, where $d \in \mathcal{D} \subset \mathbb{R}^{|S| \times |A|}$ satisfies the non-negativity constraint ($d \succeq 0$) and the flow conservation constraint: $\sum_a d(s,a) - \gamma \sum_{s',a'} d(s',a') P(s \mid s', a') = \mu(s), \quad \forall s \in \mathcal{S}$. The feasible set $\mathcal{D}$ forms a convex polytope [2], whereas the set of value functions, $\mathcal{V}$, is a polytope that is generally non-convex [6]. This dual formulation offers several advantages, including efficient handling of constraints and the ability to solve the problem using linear programming techniques.

For robust MDPs, the geometry of robust value functions is significantly more intricate compared to standard MDPs [32]. While the dual formulation for standard MDPs is well-established, this work is the first to derive a dual formulation for this specific class of non-rectangular robust MDPs, providing critical insights and laying the foundation for the development of robust policy evaluation methods.

## 3  Method

In this section, we introduce $L_p$-bounded non-rectangular uncertainty set, and demonstrate that its rectangular relaxation may yield highly sub-optimal solutions. Then, we establish that this

uncertainty set avoids the NP-Hardness results established in [36]. Subsequently, we show that the robust evaluation for this uncertainty set is equivalent to robust evaluation over an infinite collection of sa-rectangular robust MDPs. This equivalence leads to a novel dual formulation and, ultimately, a binary search method for robust policy evaluation. We begin with defining non-rectangular $L_p$-bounded uncertainty set as:

$$\mathcal{U}_p = \left\{ P \mid \|P - \hat{P}\|_p \leq \beta, \; \sum_{s'} P(s' \mid s, a) = 1 \right\},$$

where $\hat{P}$ is the nominal kernel, $\beta$ is uncertainty radius, and $\|P - \hat{P}\|_p = (\sum_{s,a,s'} (P(s'|s,a) - \hat{P}(s'|s,a))^p)^{\frac{1}{p}}$. The simplex constraint ensures that the transition kernel $P$ satisfies the unity-sum-rows property, as discussed in [20]. Following previous works [7, 20, 18], we assume the radius $\beta$ is sufficiently small to ensure all the kernels within the uncertainty sets are valid transition kernels. As discussed in [20], this assumption can be lifted by imposing boundary constraints ($0 \leq P(s|s,a) \leq 1$) at the expense of additional complexity and without yielding significant additional insights. Throughout the paper, we use $d^\pi, v^\pi, J^\pi, D^\pi$ as shorthand for $d^\pi_{\hat{P}}, v^\pi_{\hat{P}}, J^\pi_{\hat{P}}$, and $D^\pi_{\hat{P}}$, respectively, w.r.t. nominal kernel $\hat{P}$.

**Why non-rectangular RMDPs.** Note that the non-rectangular uncertainty sets allow noise in one state to be coupled with noise in other states. Before delving into solving them, we first discuss their importance. Why are uncertainty sets modeled with non-rectangular sets $\mathcal{U}_p$ (e.g., $L_2$-balls) better than rectangular ones?

In Figure 1, we illustrate this by capturing the uncertainty set using non-rectangular $\mathcal{U}_2$ (circle/sphere) balls and rectangular (square/cube) balls. The blue dots represent possible environments, with the origin being the nominal environment. Points farther away from the origin indicate larger perturbations. Specifically, points near the corners of the square/cube represent environments with large perturbations in nearly all dimensions or coordinates simultaneously. The likelihood of such simultaneous perturbations is very low, and this issue becomes even more pronounced in higher dimensions. This phenomenon is well discussed in the paper *Lightning Doesn't Strike Twice: Coupled RMDPs*[22].

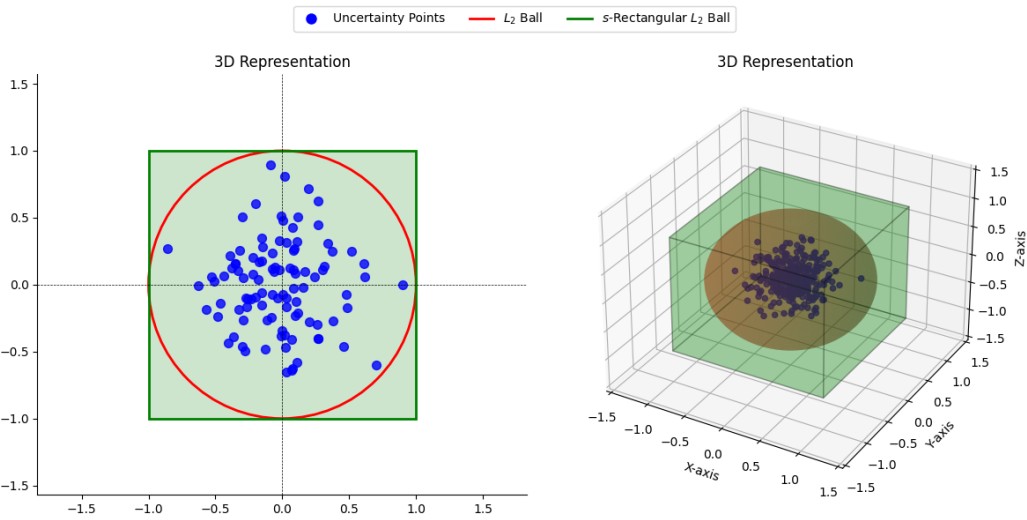

Figure 1: Modeling Uncertainty with Non-Rectangular and Rectangular $L_2$-Balls.

Moreover, as shown in the result below, most of the volume of a high-dimensional cube lies near its corners outside the embedded sphere. This implies that rectangular robust MDPs are overly conservative, as their uncertainty sets focus on environments near the corners—corresponding to highly unlikely extreme perturbations.

**Proposition 3.1.** *Let $\mathcal{U}_2^{\mathtt{sa}}$ and $\mathcal{U}_2^{\mathtt{s}}$ denote the smallest $\mathtt{sa}$-rectangular and $\mathtt{s}$-rectangular sets, respectively, that contain $\mathcal{U}_2$. Then:*

$$\frac{\boldsymbol{vol}(\mathcal{U}_2)}{\boldsymbol{vol}(\mathcal{U}_2^{\mathtt{sa}})} = O(c_{sa}^{-SA}), \quad \text{and} \quad \frac{\boldsymbol{vol}(\mathcal{U}_2)}{\boldsymbol{vol}(\mathcal{U}_2^{\mathtt{s}})} = O(c_s^{-S}),$$

*where $\boldsymbol{vol}(X)$ denotes the volume of the set $X$, $S = |\mathcal{S}|, A = |A|$ and $c_s, c_{sa} > 1$ are constants.*

The result follows from comparing the $n$-dimensional sphere's volume $c_n r^n$ ($c_n \to 0$) [28], to the enclosing cube's volume $2^n r^n$ (side $2r$), resulting in a ratio of $O(2^{-n})$.

In summary, real-world uncertainty sets are often non-rectangular and highly coupled. Their rectangular relaxations (the smallest rectangular uncertainty sets encapsulating the original non-rectangular sets) introduce exponentially more additional environments, many of which correspond to highly perturbed kernels that are improbable in practice. As a result, relaxed rectangular robust MDPs can produce overly conservative and suboptimal solutions compared to their non-rectangular counterparts.

**Complexity.** While non-rectangular robust MDPs better capture real-world uncertainty sets, robust policy evaluation (even approximation) has been proven NP-hard for general uncertainty sets defined as intersections of finite hyperplanes [36]. Specifically, [36] reduces an Integer Program (IP) with $m$ constraints to robust MDPs where the uncertainty set consists of intersections of $m$ half-spaces ($m$-linear constraints). This polyhedral structure is fundamental to the hardness proof, consequently, it does not extend to our uncertainty sets $\mathcal{U}_p$ for $p > 1$. For the case of $\mathcal{U}_1$, the IP reduction does apply, but since $\mathcal{U}_1$ is defined by a single global constraint ($\|P - \hat{P}_1\|_1 \leq \beta$), this implies that the corresponding IP has only one simple constraint which is efficiently solvable. A detailed discussion can be found in Appendix B.2.

Intuitively, the NP-Hardness result primarily applies to polyhedral uncertainty sets with numerous vertices. This leaves room for the possibility that many uncertainty sets defined by a small number of global constraints, such as norms or distances, might fall outside the scope of this hardness and could potentially be tractable. However, we leave this intriguing question for future exploration: is NP-Hardness merely the tip of the iceberg?

**Divide and Conquer.** The above discussion highlights the potential tractability of $L_p$-robust MDPs, prompting us to address the challenge of solving them. A key insight is that the non-rectangular uncertainty set $\mathcal{U}_p$ can be expressed as a union of $\mathtt{sa}$-rectangular sets $\mathcal{U}_p^{\mathtt{sa}}(b)$ with varying radius vectors $b$, as formalized in the result below. Each $\mathtt{sa}$-rectangular set can be efficiently solved individually, paving the way for a more manageable approach to the overall problem.

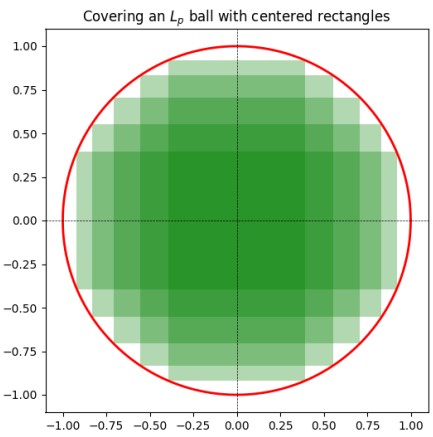

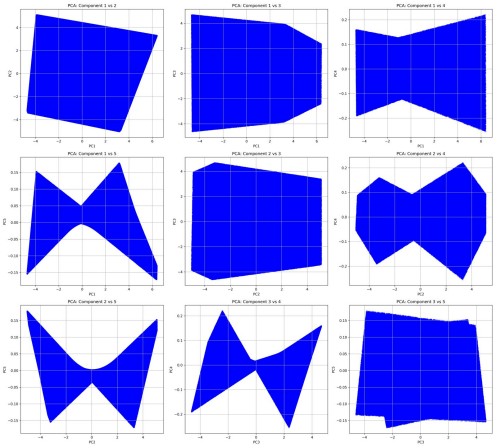

(a) Illustration of Proposition 3.2: N-dimensional sphere can be written as infinite union of n-dimensional inscribing cubes.

(b) Projections of set $\mathcal{D}$ along principal components, for $S = 3, A = 2$ with 10 millions samples, strongly suggesting non-convexity.

Figure 2:

**Proposition 3.2.** *[Decomposition] The non-rectangular uncertainty set $\mathcal{U}_p$ can be expressed as:*

$$\mathcal{U}_p = \bigcup_{b \in \mathcal{B}} \mathcal{U}_p^{sa}(b),$$

*where $\mathcal{B} = \{b \in \mathbb{R}_+^{\mathcal{S} \times \mathcal{A}} \mid \|b\|_p \leq \beta\}$, and $\mathcal{U}_p^{sa}(b) = \{P \mid \|P(\cdot \mid s,a) - \hat{P}(\cdot \mid s,a)\|_p \leq b(s,a), \forall(s,a)\}$, is $\mathit{sa}$-rectangular uncertainty set with radius vector $b$.*

The proof of the above result intuitively generalizes the idea that a circle (or $n$-dimensional sphere) can be covered by an inscribed square (or $n$-dimensional rectangles) touching its boundaries and a continuum of its rotated versions, as shown in Figure 2(a). This offers a significant simplification to the problem at hand, as it implies that non-rectangular policy evaluation (difficult) can be decomposed into $\mathtt{sa}$-rectangular uncertainty sets (easier) as:

$$J_{\mathcal{U}_p}^\pi = \min_{b \in \mathcal{B}} \min_{P \in \mathcal{U}_p^{\mathtt{sa}}(b)} J_P^\pi. \tag{1}$$

In essence, we have simplified a complex problem into an infinite number of more manageable ones. However, the task remains incomplete. Although a closed-form expression exists for $J_{\mathcal{U}_p^{\mathtt{sa}}}^\pi = J^\pi - \sum_{s,a} d^\pi(s,a) b_{sa} \sigma_q(v_{\mathcal{U}_p^{\mathtt{sa}}}^\pi)$, where $q$ is the Hölder conjugate of $p$ (i.e., $\frac{1}{p} + \frac{1}{q} = 1$) and $\sigma_p$ is the generalized standard deviation (GSTD) defined as $\sigma_p(v) = \min_{\omega \in \mathbb{R}} \|v - \omega \mathbf{1}\|_p$ [20], this approach is still computationally impractical. The core challenge lies in solving $\max_{b \in \mathcal{B}} \sum_{s,a} d^\pi(s,a) b_{sa} \sigma_q(v_{\mathcal{U}_p^{\mathtt{sa}}(b)}^\pi)$, which remains a formidable task. To circumvent this, we leverage the dual formalism, which is elaborated in the next section.

### 3.1 Dual Formulation of Robust MDPs

Here, we present a dual formulation for robust MDPs specifically for $L_p$-bounded uncertainty sets. While this formulation is inherently more intricate than the classical dual formulation for standard MDPs [27], it forms the foundation for all subsequent results in this work.

Now, leveraging results from [18], we know that the worst-case kernel for $\mathtt{sa}$-rectangular uncertainty sets, $P_{\mathcal{U}_p^{\mathtt{sa}}(b)}^\pi = \hat{P} - bk^\top$, can be expressed as a rank-one perturbation of the nominal kernel, where $k \in \mathcal{K} := \{k \mid \|k\|_p \leq 1, \mathbf{1}^\top k = 0\}$. Consequently, the adversary can restrict their focus to rank-one perturbations, enabling us to reformulate the robust return as:

$$J_{\mathcal{U}_p}^\pi = \min_{b \in \mathcal{B}} \min_{k \in \mathcal{K}} J_{\hat{P} - bk^\top}^\pi = \min_{b \in \mathcal{B}} \min_{k \in \mathcal{K}} \mu^\top D_{\hat{P} - bk^\top}^\pi R^\pi,$$

where the last equality stems from $J_P^\pi = \mu^\top D_P^\pi R^\pi$. Further, leveraging Lemma 4.4 from [18] or directly applying the Sherman–Morrison formula [4] (see Proposition D.1), the robust return can be expressed as:

$$J_{\mathcal{U}_p}^\pi = \min_{b \in \mathcal{B}, k \in \mathcal{K}} \left[ \mu^\top D^\pi R^\pi - \gamma \mu^\top D^\pi b^\pi \frac{k^\top D^\pi R^\pi}{1 + \gamma k^\top D^\pi b^\pi} \right],$$

where $b_s^\pi := \sum_a \pi(a|s) b_{sa}$. The following result introduces a more concise and interpretable form of this robust return expression.

**Lemma 3.3.** *[Penalized Robust Return] The robust return can be expressed as:*

$$J_{\mathcal{U}_p}^\pi = J^\pi - \gamma \max_{b \in \mathcal{B}, k \in \mathcal{K}} \frac{\langle k, v_R^\pi \rangle \langle d^\pi, b^\pi \rangle}{1 + \gamma \langle k, v_b^\pi \rangle},$$

*where $v_b^\pi = D^\pi b^\pi$ represents the value function with uncertainty radius $b$ as the reward vector.*

For the first time, the above result expresses the robust return in terms of the nominal return $J^\pi$ and a penalty term involving only nominal values ($d^\pi$, $v_R^\pi = v^\pi$, and $v_b^\pi$). Notably, the denominator term $1 + \gamma \langle k, v_b^\pi \rangle$ is strictly positive (see appendix for details). In the subsequent subsections, we delve deeper into evaluating this penalty term and analyzing the nature of the optimal $(k, b)$ for a given policy $\pi$, revealing the adversary. Finally, by maximizing the robust return $J_{\mathcal{U}_p}^\pi$ over policies, we get a dual formulation, as stated below.

**Theorem 3.4** (Dual Formulation). *The optimal robust return is the solution to*

$$J_{\mathcal{U}_p}^* = \max_{D \in \mathcal{D}} \min_{k \in \mathcal{K}, b \in \mathcal{B}} \left[ \mu^T D R - \gamma \mu^T D b \frac{k^T D R}{1 + \gamma k^T D b} \right]$$

*where* $\mathcal{D} = \{ D^\pi H^\pi \mid \pi \in \Pi \}$, $D^\pi = (I - \gamma \hat{P}^\pi)^{-1}$ *and* $H^\pi : \mathbb{R}^{\mathcal{S} \times \mathcal{A}} \to \mathbb{R}^{\mathcal{S}}$ *is a policy averaging linear operator defined as* $H^\pi R := R^\pi$.

The dual formulations for the `sa`-rectangular and `s`-rectangular cases differ notably in their definitions of $\mathcal{B}$. In the `sa`-rectangular case, $\mathcal{B} = \{\beta\}$, whereas in the `s`-rectangular case, $\mathcal{B} = \{b \in \mathbb{R}^{\mathcal{S} \times \mathcal{A}} \mid \|b_s\|_p \leq \beta_s\}$. These distinctions are elaborated in the appendix. The result above frames the dual of robust MDPs as a min-max problem, offering valuable and insightful perspectives. However, as Figure 2(b) suggests (with further details in the appendix), the set $\mathcal{D}$ may be non-convex, which complicates the problem. Despite this, we believe that the dual formulation holds potential for future work, providing deeper insights and enabling the development of improved algorithms. In this work, we keep our focus on the robust policy evaluation while policy improvement is addressed via existing robust policy gradient method with proven guarantees [33], discussed further in Appendix C.

## 3.2 Robust Policy Evaluation

Now, we directly attempt to evaluate the penalty term in Lemma 3.3 which leads to a binary search-based robust policy evaluation algorithm. The key idea is to identify a bisection function:

$$F(\lambda) = \max_{b \in \mathcal{B}, k \in \mathcal{K}} k^\top E_\lambda^\pi b,$$

where $E_\lambda^\pi := \gamma \left[ D^\pi R^\pi \mu^\top D^\pi - \lambda D^\pi \right] H^\pi$. Note that $E_\lambda^\pi$ is constructed using quantities that are computationally straightforward, and $H^\pi : \mathbb{R}^{\mathcal{S} \times \mathcal{A}} \to \mathbb{R}^{\mathcal{S}}$ represents the policy-averaging linear operator, defined by $(H^\pi R)(s) := \sum_a \pi(a|s) R(s, a)$.

**Proposition 3.5.** *For* $p = 1$, $F$ *can be evaluated in closed form as*

$$F(\lambda) = \gamma \beta \max_{s \in \mathcal{S}, a \in \mathcal{A}} \pi(a|s) \|d^\pi(s) v^\pi - \lambda D^\pi(\cdot, s)\|_{sp},$$

*where* $\|x\|_{sp} = \frac{\max_i x_i - \min_i x_i}{2}$ *and* $D^\pi = (I - \gamma \hat{P}^\pi)^{-1}$.

*Proof.* The proof is referred to Proposition G.4. $\qquad \square$

Note that the above result requires computation of $v^\pi, d^\pi, D^\pi$ which can be approximated iteratively with time complexity $O(S^2 A \log \epsilon^{-1})$ or exactly via matrix inversion with time complexity $O(S^3 A^3)$.

**Lemma 3.6** (Robust Policy Evaluation). *Let* $\lambda^*$ *be a fixed point of the function* $F(\lambda)$, *then the robust return can be expressed as:*

$$J_{\mathcal{U}_p}^\pi = J^\pi - \lambda^*.$$

*And* $\lambda^*$ *can be efficiently computed using binary search Algorithm 1 as* $F(\lambda) > \lambda \iff \lambda > \lambda^*$.

*Proof.* The proof can be found in Appendix (see Lemma F.2). $\qquad \square$

The result enables a direct computation of the robust return by iteratively refining $\lambda$ until convergence, leveraging the monotonicity properties of $F(\lambda)$. Further, the bisection property of $F$ established in the result, directly implies the linear convergence rate of Algorithm 1, as stated result below.

---

**Algorithm 1** Binary Search for Robust Policy Evaluation

---

**Initialize:** Upper limit $\lambda_u = \frac{1}{1-\gamma}$, lower limit $\lambda_l = 0$

1: **while** not converged: $n = n + 1$ **do**
2:    **Bisection value:** $\lambda_n = (\lambda_l + \lambda_u)/2$
3:    **Bisection:** $\lambda_l = \lambda_n$ if $F(\lambda_n) > \lambda_n$, else $\lambda_u = \lambda_n$.
4:    **Update robust return:** $J_n = J_P^\pi$ where $P = \mathbf{proj}(P_0 - bk^T)$,
     $(b, k) \in \arg\max_{b \in \mathcal{B}, k \in \mathcal{K}} k^\top E_\lambda^\pi b$, and **proj** is simplex projection operator.
5: **end while**

---

**Theorem 3.7.** *Algorithm 1 converges linearly, i.e.,*

$$J_n - J_{\mathcal{U}_p}^\pi \leq O(2^{-n}).$$

We conclude that robust evaluation can be performed efficiently with linear iteration complexity. However, each iteration involves solving the subproblem $\max_{x \in \mathcal{B}} \|Ax\|_q$, as part of Algorithm 1 which can solved in closed form shown in Proposition 3.5. For $p = 2$, we provide a spectral heuristic for its approximation in the appendix, however for other $p \neq 1, 2$, it can be hard.

## 4 Revealing the Adversary

We provide the first insights into the structure of the worst-case kernel in non-rectangular robust MDPs, addressing an unexplored area in the literature. The following result reveals that, similar to rectangular uncertainty sets [18], the worst-case transition kernel is a rank-one perturbation of the nominal kernel, but with a more complex structure.

**Theorem 4.1** (Worst-Case Kernel). *For a policy $\pi$ and uncertainty set $\mathcal{U}_p$, the worst-case transition kernel is:*

$$P_{\mathcal{U}_p}^\pi = \hat{P} - bk^\top,$$

*where $(k, b)$ solves:*

$$\max_{k \in \mathcal{K}, \, b \in \mathcal{B}} \frac{J_b^\pi \langle k, v_R^\pi \rangle}{1 + \gamma \langle k, v_b^\pi \rangle}.$$

The above result follows directly from Lemma 3.3. It highlights the adversary's strategic use of $k$, $b$, and their interaction with the value functions $v_R^\pi$ and $v_b^\pi$, revealing a more nuanced structure compared to the rectangular case. The adversary's objectives in selecting the worst-case kernel are twofold:

- **Maximizing Trajectory Uncertainty** ($J_b^\pi$): The adversary seeks to increase the agent's visits to high-uncertainty states, enhancing its ability to steer the agent toward disadvantageous outcomes.

- **Optimizing the Perturbation Direction** ($k$): The adversary selects $k$ to maximize $k^\top v_R^\pi$, thereby pushing the agent into low-reward trajectories, while simultaneously minimizing $k^\top v_b^\pi$ to ensure the agent remains exposed to high-uncertainty states.

These insights provide a deeper understanding of the adversary's behavior and offer practical guidance for designing more resilient robust algorithms to counteract such strategies effectively.

---

**Message to Practitioners**

The adversary focuses solely on rank-one perturbations of the nominal kernel, iteratively boosting its influence by pushing the agent into high-uncertainty states, then leveraging that influence to steer the agent toward low-reward trajectories, ultimately driving the agent to the lowest possible return.

---

## 5 Experiments: Robust Policy Evaluation with $L_1$ Normed Uncertainty Set

We conduct a numerical comparison of our Algorithm 1 and CPI (Algorithm 3.2 from [21], reproduced as Algorithm 2 in the appendix) for robust policy evaluation. The experiments are performed using a randomly generated nominal kernel $\hat{P}$, reward function $R$, and policy $\pi$. An uncertainty set $\mathcal{U}_1$ is constructed using the nominal kernel with a fixed uncertainty radius $\beta$.

Figure 14 demonstrates the convergence behavior of both methods, presenting results based on the number of iterations (left panel) and computation time (right panel). The left panel shows the robust return achieved per iteration, while the right panel depicts the robust return as a function of wall-clock time. Note that the x-axes of the figure have a logarithmic scale in order to clearly capture the slow convergence of the CPI method.

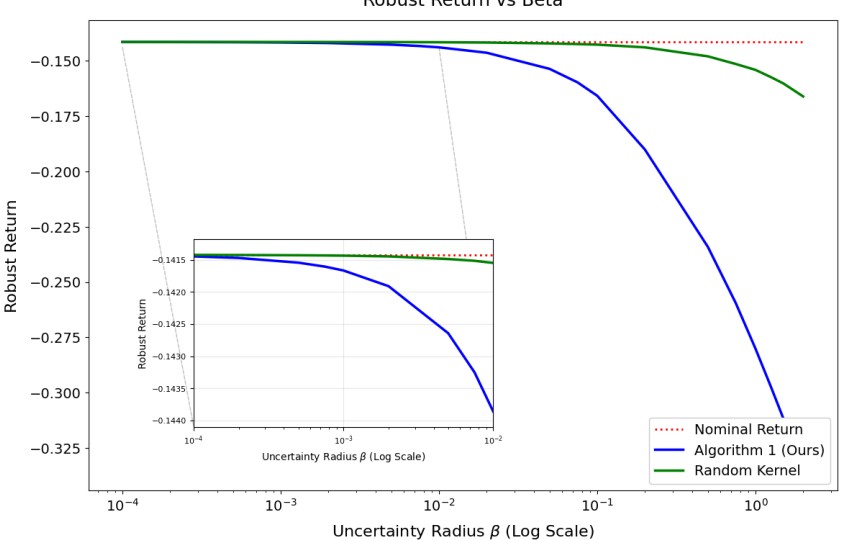

Figure 3: Robust Return for $\beta$ (Uncertainty Radius, S=12 and A=8, $\gamma$=0.9 and convergence tolerance of $10^{-4}$ . Robust Return of Random Kernel is minimum of 500 millions kernels samples for each $\beta$

- **Our Algorithm 1.** We apply our Binary Search Algorithm 1 to perform robust policy evaluation with the given nominal kernel $\hat{P}$ and uncertainty radius $\beta$. Each iteration of the algorithm involves computing $F(\lambda)$, using close form solution provided in Proposition 3.5. Our algorithm converges very quickly requiring only a few iterations.

- **Algorithm 3.2 of [21].** We run Algorithm 2 with precomputed values of $d^\pi$ and $A^\pi$. The step sizes are chosen to be either a small constant or dynamically adjusted, as described in the algorithm. Note that Line 3 of the algorithm involves solving $\arg\min_{P \in \mathcal{U}_1}\langle x, P\rangle$. This constrained optimization is solved using a numerical method (scipy.minimize). This gradient based method improves very slowly and converges very far from the true robust return as the uncertainty set $\mathcal{U}_1$ is very non-rectangular.

- **Brute Force Benchmark.** To approximate the true robust return, we generate a large number of random samples $\{P_i \mid i \leq n\}$ from $\mathcal{U}_1$ and estimate the empirical minimum, $\min_i J^\pi_{P_i}$, as a proxy for the robust return. Note this method requires exponential number of samples to reasonably cover the entire uncertainty set. Hence the values obtained in Figure 14, are an approximate upper bound on the true robust return.

The results in Figure 14 reflect a general trend observed across a wide range of experiments conducted with state space sizes ranging from $S = 5$ to $S = 190$ and uncertainty radius $\beta \in \{0.005, 0.01, 0.05\}$. Our proposed algorithm consistently demonstrates superior performance, converging in significantly fewer iterations and less computation time while the computational demands of the CPI algorithm grow substantially with larger state spaces. Hence, our method exhibits more favorable scaling properties, making it practical for high-dimensional problems.

The codes, detailed explanations, and additional experiments are available at `https://anonymous.4open.science/r/non-rectangular-rmdp-77B8`. System details for the experiments are as follows: **Operating System:** macOS Sequoia (Version 15.4.1), **Chip:** Apple M2, **Cores:** 8 (4 performance and 4 efficiency), **Memory:** 16 GB (LPDDR5).

## 6 Discussion

We studied robust Markov decision processes (RMDPs) with non-rectangular $L_p$-bounded uncertainty sets, balancing expressiveness and tractability. We showed that these uncertainty sets can be decomposed into infinitely many $sa$-rectangular sets, reducing robust policy evaluation to a min-max fractional optimization problem (dual form). This novel dual formulation provides key insights

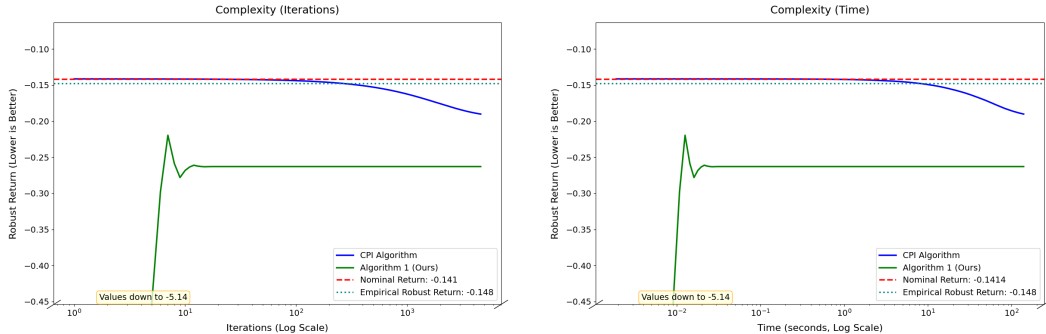

Figure 4: Comparison of Algorithm 1 (Ours) and the CPI Algorithm for $\beta$ (Uncertainty Radius) $= 0.05$, $S = 10$, $A = 8$, $\gamma = 0.9$, and a convergence tolerance of $10^{-4}$.

into the adversary and leads to the development of an efficient robust policy evaluation algorithm. Particularly, we provided first polynomial time policy evaluation for non-rectangular $L_1$ normed uncertainty set. These findings further pave the way for scalable and efficient robust reinforcement learning algorithms.

**Limitations.** Similar to [7, 20, 18], we have considered small enough uncertainty radius to ensure positivity of the kernel. As discussed in [20], imposing this additional positivity constraints (or dealing with nominal kernel with zero transition probability to some states ) would significantly complicate the analysis without yielding significant additional insights. However, we leave a thorough investigation of this topic for future work.

Optimal robust policies for non-rectangular uncertainty sets may be history dependent [36], but we focus on stationary policies since even their evaluation is NP-hard [36]. Extending to non-stationary cases is left for future work.

**Future Work.** Our results naturally extend to uncertainty sets that can be expressed as a finite union of $L_p$ balls. Furthermore, any uncertainty set can be approximated using a finite number of $L_p$ balls, with smaller balls providing a better approximation. However, the number of balls required for an accurate approximation may grow prohibitively large. While this work is limited to $L_p$ norms, it may be possible to generalize our approach to other types of uncertainty sets. A key challenge in such an extension would be identifying the structure of the worst-case kernel and developing corresponding matrix inversion techniques.

Particularly, in the case of KL-divergence–based uncertainty sets, the structure of the worst-case transition kernel is known [31]—it takes an exponentiated form rather than a rank-one form. It may be possible to derive an analogue of the Sherman-Morrison formula, or a practically useful approximation, for this case.

## Acknowledgments and Disclosure of Funding

This research was partially supported by Israel PBC- VATAT, by the Technion Artificial Intelligent Hub (Tech.AI) and by the Israel Science Foundation (grant No. 447/20).

Additionally, part of this work was supported by the Israel Science Foundation (grant No. 3019/24).

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

## Notations and Definitions

For a set $\mathcal{S}$, $|\mathcal{S}|$ denotes its cardinality. $\langle u, v \rangle := \sum_{s \in \mathcal{S}} u(s)v(s)$ denotes the dot product between functions $u, v : \mathcal{S} \to \mathbb{R}$. $\|v\|_p^q := (\sum_s |v(s)|^p)^{\frac{q}{p}}$ denotes the $q$-th power of $L_p$ norm of function $v$, and we use $\|v\|_p := \|v\|_p^1$ and $\|v\| := \|v\|_2$ as shorthand. For a set $\mathcal{C}$, $\Delta_{\mathcal{C}} := \{a : \mathcal{C} \to \mathbb{R} | a_c \geq 0, \forall c, \sum_{c \in \mathcal{C}} a_c = 1\}$ is the probability simplex over $\mathcal{C}$. $\text{var}(\cdot)$ is variance function, defined as $\text{var}(v) = \sqrt{\sum_{s \in \mathcal{S}} (v(s) - \bar{v})^2}$ where $\bar{v} = \frac{\sum_{s \in \mathcal{S}} v(s)}{|\mathcal{S}|}$ is the mean of function $v : \mathcal{S} \to \mathbb{R}^d$. $\mathbf{0}, \mathbf{1}$ denotes all zero vector and all ones vector/function respectively of appropriate dimension/domain. $\mathbf{1}(a = b) := 1$ if $a = b$, 0 otherwise, is the indicator function. For vectors $u, v$, $\mathbf{1}(u \geq v)$ is component wise indicator vector, i.e. $\mathbf{1}(u \geq v)(x) = \mathbf{1}(u(x) \geq v(x))$. $A \times B = \{(a, b) \mid a \in A, b \in B\}$ is the Cartesian product between set $A$ and $B$.

## A  Related Work

**Rectangular Robust MDPs.** In the literature, the sa-rectangular uncertainty is a very old assumption [16, 25]. [36] introduced s-rectangular uncertainty sets and proved its tractability, in addition to the intractability of the general non-rectangular uncertainty sets. The most advantageous aspect of the s-rectangularity, is the existence of contractive robust Bellman operators. This gave rise to many robust value based methods [14, 33]. Further, for many specific uncertainty sets, robust Bellman operators are equivalent to regularized non-robust operators, making the robust value iteration as efficient as non-robust MDPs [7, 34, 20]. There exists many policy gradient based methods for robust

Table 2: Useful Notations

| Notation | Definition | Remark |
|---|---|---|
| $p, q$ | $\frac{1}{p} + \frac{1}{q} = 1$ | Holder's conjugates |
| $\sigma_p$ | | Standard deviation w.r.t. $L_p$ norm |
| $v^\pi, v^\pi_{P,R}$ | $(I - \gamma P^\pi)^{-1} R^\pi$ | Value function |
| $D^\pi, D^\pi_{P,R}$ | $(I - \gamma P^\pi)^{-1}$ | Occupancy matrix |
| $d^\pi, d^\pi_{P,\mu}$ | $\mu^T(I - \gamma P^\pi)^{-1}$ | Occupancy measure |
| $\mathcal{U}, \mathcal{U}^{\mathsf{sa}}_p, \mathcal{U}^{\mathsf{s}}_p, \mathcal{U}_p$ | | Uncertainty sets |

MDPs, relying upon contractive robust Bellman operators for the robust policy evaluation [35, 18]. Further, [39, 31] try to refine the process, and directly get samples from the adversarial model via pessimistic sampling . There exist other notions of rectangularity such as k-rectangularity [23] and r-rectangularity [10] which are sparsely studied. However, [11] shows, the theses uncertainty sets are either equivalent to s-rectangularity or non-tractable.

**Non-Rectangular Reward Robust MDPs.** Policy evaluation for robust MDPs with non-rectangular uncertainty set is proven to be a Strongly-NP-Hard problem [36], in general. For a very specific case, where uncertainty is limited only to reward uncertainty bounded with $L_p$ norm, [8] proposed robust policy evaluation via frequency (occupation measure) regularization, and derived the policy gradient for policy improvement.

**Approximate Policy Evaluation for Non-Rectangular Kernel RMDPs.** [21] provides the following two policy evaluation methods for robust MDPs for general uncertainty sets.

- Langevian dynamics based Algorithm 3.1 of [21]: This Langevian dynamics based Markov Chain Monte Carlo method solves the robust policy evaluation problem to global optimality with arbitrary small accuracy $\epsilon$. The iteration complexity of the algorithm is $O(2^q \log \frac{1}{\epsilon})$ which is exponential in the dimension of the uncertainty set $q$. The algorithm is well suited only for small dimensional uncertainty. For a general case the dimension $q = S^2 A$ can be very large, this makes the algorithm very computationally inefficient as expected from the hardness result from [2].

- CPI sytle Algorithm 3.2 of [21] (presented in Algorithm 2): This CPI based algorithm computes the robust policy with iteration complexity of $O(\frac{1}{\epsilon^2})$ with an accuracy of $\delta_d(2\epsilon + \delta_{\mathcal{P}})$, where $\delta_d$ is mismatch-coefficient and $\delta_{\mathcal{P}}$ is measure of non-rectangularity of the uncertainty set. However, the mismatch coefficient may not exist without an irreducibility assumption (Assumption 1 in [21]), moreover even under Assumption 1, the constant $\delta_d = O(2^S)$ can be exponentially large for ladder MDPs which have large diameter (more details provided below). In addition, the non-rectangularity constant $\delta_{\mathcal{P}}$ can be as large as $O(\sqrt{S})$. Hence, a large $\delta_d \delta_{\mathcal{P}} > \frac{2}{1-\gamma}$ makes the bound meaningless, as the sub-optimality is upper bounded by $\frac{2}{1-\gamma}$. To summarize, this approach is efficient only for small diameter MDPs and almost rectangular uncertianty sets.

- Our Method: We provide a robust policy evaluation method for $L_2$-robust MDPs with an iteration complexity of $O(\log \frac{1}{\epsilon})$ and with an accuracy of $\epsilon$. This is possible as we showed that the NP-hardness result of [2] doesn't apply to this case.

  We don't require the irreducibility Assumption 1 of [21] which can be very limiting. Further, the $L_p$ robust MDPs may have very large tolerance $\delta_d \delta_{\mathcal{P}}$ hence the Algorithm 3.2 from [21] is not applicable.

**Difficult MDPs for Algorithm 3.2 of [21]** :

- **MDP with high mismatch coefficients** : Consider an MDP with only one action and state-space $\{s_i | 1 \le i \le S\}$. Let $s_1$ be the starting state. Let the kernel be defined as

$$P_x(s_{\max\{i+1,S\}} | s_i) = x, \qquad P_x(s_i | s_i) = 1 - x.$$

Now let the uncertainty set be $\mathcal{P} = \{P_x \mid x \in [0.4, 0.6]\}$. Note that for this case, $\log(\delta_d) \ge \log(\frac{d^{P_{0.6}}(s_S | s_1)}{d^{P_{0.4}}(s_S | s_1)}) = O(S)$.

- **High non-rectangularity coefficient** : This is inspired from the fact that

$$\delta = \max_{||a|| \le 1} \left[ \max_{b \in \mathcal{B}_f} < a, b > - \max_{b \in \mathcal{B}} < a, b > \right],$$

where $\mathcal{B} = B(0, 1)$ is a unit ball around origin, and $\mathcal{B}_f = [-1, 1]^n$ is the smallest rectangular cube containing $\mathcal{B}$. Then choosing $a = \{\frac{1}{\sqrt{n}}\}^n$, we have $\max_{b \in \mathcal{B}_f} < a, b >= \sqrt{n}$ and $\max_{b \in \mathcal{B}} < a, b >= 1$. This implies $\delta \ge \sqrt{n} - 1$.

From definition in page 11 of [21], we have

$$\delta_{\mathcal{P}} = \max_{P \in \mathcal{P}_f} < \nabla V, P > - \max_{P \in \mathcal{P}} < \nabla V, P >$$

where $\mathcal{P}_f$ is the smallest s-rectangular uncertainty containing $\mathcal{P}$. Here, $P \in \mathbb{R}^{SA \times S}$, this suggests $\delta_{\mathcal{P}}$ can be of the order of $O(S\sqrt{A})$.

The discussion is summarize in the Table 1.

---

**Algorithm 2** CPI Algorithm 3.2 of [21] for Robust Policy Evaluation

---

**Input:** Nominal kernel $\hat{P}$, policy $\pi$, Uncertainty set $\mathcal{U}$.

1: **while** not converged: $n = n + 1$ **do**
2:   Define : $f(P) := \frac{1}{1-\gamma} \sum_{s,a,s'} d_{\hat{P}}^{\pi}(s) \pi(a|s) A_{\hat{P}}^{\pi}(s, a, s') P(s'|s, a)$,

   where $A_P^{\pi}(s, a, s') := \gamma \left[ P(s'|s, a) v_P^{\pi}(s') - \sum_{s''} P(s''|s, a) v_P^{\pi}(s') \right]$.

3:   Compute $P^* \in \arg\min_{P \in \mathcal{U}} f(P)$.

4:   Update the estimated worst kernel: $P_{n+1} = (1 - \alpha_n) P_n + \alpha_n P^*$,
   where $\alpha_n = -\frac{(1-\gamma)^3}{4\gamma^2} f(P^*)$
5: **end while**
**Return:** Robust return $J_{P_\infty}^{\pi}$.

---

**Robust Policy Gradient Methods.** The absence of contractive robust Bellman operators renders the development of value-based methods for robust MDPs particularly challenging. Consequently, policy gradient methods naturally emerge as a viable alternative. The update rule is given by:

$$\pi_{k+1} = \text{Proj}_{\pi \in \Pi} \left[ \pi_k - \eta_k \nabla_\pi J_{P_k}^{\pi_k} \right], \tag{2}$$

where $J_{P_k}^{\pi_k} - J_{\mathcal{U}}^{\pi_k} \le \epsilon \gamma^k$ and learning rate $\eta_k = O(\frac{1}{\sqrt{k}})$. This approach guarantees convergence to a global solution within $O(\epsilon^{-4})$ iterations [33].

However, this update rule depends on oracle access to the robust gradient, which is highly challenging to obtain because robust policy evaluation is an NP-hard problem.

# B  On the Non-Rectangular Uncertainty Sets

## B.1  Why non-rectangular RMPDs

**Proposition B.1.** *Let $\mathcal{U}_2^{sa}, \mathcal{U}_2^s$ be the smallest sa-rectangular set and s-rectangular set containing $\mathcal{U}_2$ then*

$$\frac{\boldsymbol{vol}(\mathcal{U}_2)}{\boldsymbol{vol}(\mathcal{U}_2^{sa})} = O(c_{sa}^{-SA}), \quad \text{and} \quad \frac{\boldsymbol{vol}(\mathcal{U}_2)}{\boldsymbol{vol}(\mathcal{U}_2^s)} = O(c_s^{-S}),$$

*where $\boldsymbol{vol}(X)$ is volume of the set $X$ and $c_s, c_{sa} > 1$ are some constants.*

*Proof.* Volume of n-dimension sphere of radius $r$ is $c_n r^n$ where $c_n \leq \frac{8\pi^2}{15}$ [28]. And to cover an n-dimension sphere of radius $r$, we need a cube of radius $2r$ whose volume is $(2r)^n$. Hence the first result $\frac{\mathbf{vol}(\mathcal{U}_2)}{\mathbf{vol}(\mathcal{U}_2^{\mathrm{sa}})} = O(2^{-SA})$ immediately follows.

Now, the volume of the set of $X = \times_{s \in \mathcal{S}} X_s$ where $X_s$ is an $A$-dimension sphere of radius $r$, then the volume of $X$ is $(c_A r)^S$. And the volume of an $SA$ dimensional sphere is $c_{SA} r^{SA}$, where $\lim_{n \to \infty} c_n \to 0$ [28]. Hence the ratio of their volume is $O((c_A)^S)$, implying the other result. $\square$

## B.2 Complexity

---

### Reduction of Integer Program to Robust MDP

**0/1 Integer Program (IP):** For $g, c \in \mathcal{Z}^n, \zeta \in \mathcal{Z}, F \in \mathcal{Z}^{m \times n}$,

$$\exists x \in \{0,1\}^n \quad s.t. \quad Fx \leq g \quad \text{and} \quad c^\top x \leq \zeta?$$

is a NP-Hard problem [9], [36] which reduces into the following robust MDP.

**Robust MDP:**

1. State Space $\mathcal{S} = \{b_j, b_j^0, b_j^1 \mid j = 1, \cdots, n\} \cup \{c_0, \tau\}$, where $\tau$ is a terminal state.
2. Singleton Action Space: $\mathcal{A} = \{a\}$.
3. Uncertainty set: $\mathcal{U} = \{P_\xi \mid \xi \in [0,1]^n, F\xi \leq g\}$
4. Discount factor $\gamma \in [0,1)$; Uniform initial state distribution $\mu$.
5. Big reward $M \geq \frac{\gamma An \sum_i c_i}{2\epsilon^2}$ where $\epsilon << 1$ helps in rounding.
6. Transitions and rewards are illustrated in Figure 5

---

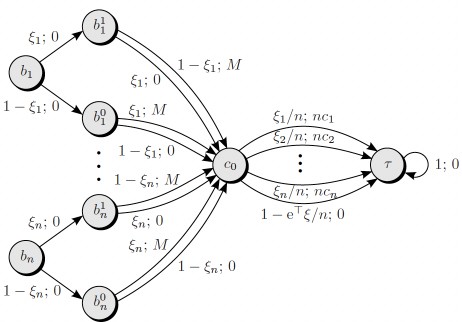

Figure 5: MDP $P_\xi$, and $R$(Figure 5 of [36]).

Robust policy evaluation is proven to be NP-hard for general uncertainty sets defined as intersections of finite hyperplanes [36]. Specifically, robust MDPs with uncertainty set $\mathcal{U}_{hard} := \{P_\xi | F\xi \leq g, \xi \in [0,1]^n\}$ where $P_\xi$ is a specially designed kernel with ladder structure with only action (effectively no decision) and a terminal state [36].

Note that $F\xi \leq g$ imposes $m$-linear constraints on $\mathcal{U}_{hard}$ while we allow only one global constraint on $\mathcal{U}_p$. Observe that $\mathcal{U}_1 = \{P_\xi \mid \mathbf{1}^\top \xi \leq g, \xi \in [0,1]^n\}$ is the nearest uncertainty to $\mathcal{U}_{hard}$ as both have polyhedral structure. This restricts the class of the IP programs to have a number of constraints $m = 1$ and the row of $F$ to be all ones. In other words, only IP programmes that can be reduced to $\mathcal{U}_1$ are of the following form: For $, c \in \mathcal{Z}^n, \zeta \in \mathcal{Z}$,

$$\exists x \in \{0,1\}^n \quad s.t. \quad \mathbf{1}^T x \leq g, \quad \text{and} \quad c^T x \leq \zeta?$$

**Solution:**

- Case 1) If $g < 0$ then **no**.
- Case 2) If $g = 0, \zeta \geq 0$ then **yes** and $g = 0, \zeta < 0$ then **yes**.

- If $g > 0$ then compute the sum of $g$ smallest coordinates of $c$, and this sum is less/equal than $\zeta$ then answer is **yes**, otherwise **no**.

Further, for IP to be reducable to robust MDPs, the diameter of the uncertainty ($\max_{P,P' \in \mathcal{U}_{hard}} \|P - P'\|_1 = 2S$) has to be large for the practical settings. Loosely speaking, robust MDPs with a $\mathcal{U}_p$ uncertainty have one global constraint and a small radius $\beta$, which corresponds to a Knapsack Problem with a small budget (IP with one constraint and a small $g$) which are much easier to solve [5, 9].

We can thus conclude that the hardness result of [36] doesn't apply to our uncertainty case.

### B.3 Decomposition

**Proposition B.2.** *Non-rectangular uncertainty $\mathcal{U}_p$ can be written as an infinite union of sa-rectangular sets $\mathcal{U}_p^{sa}$, as*

$$\mathcal{U}_p = \bigcup_{b \in \mathcal{B}} \mathcal{U}_p^{sa}(b),$$

*where $\mathcal{B} = \{b \in \mathbb{R}_+^{\mathcal{S} \times \mathcal{A}} \mid \|b\|_p \leq \beta\}$. Note that all of them share the nominal kernel $\hat{P}$.*

*Proof.* By definition, we have

$$\mathcal{U}_p = \{P \mid \|P - \hat{P}\|_p \leq \beta, \sum_{s'} P(s'|s,a) = 1\} \tag{3}$$

$$= \{P \mid \sum_{s,a} \|P_{sa} - \hat{P}_{sa}\|_p^p \leq \beta^p, \sum_{s'} P(s'|s,a) = 1\} \tag{4}$$

$$= \{P \mid \sum_{s,a} b_{sa}^p \leq \beta^p, \|P_{sa} - \hat{P}_{sa}\|_p^p = b_{sa}^p, \sum_{s'} P(s'|s,a) = 1\} \tag{5}$$

$$= \{P \mid \sum_{s,a} b_{sa}^p \leq \beta^p, \|P_{sa} - \hat{P}_{sa}\|_p^p \leq b_{sa}^p, \sum_{s'} P(s'|s,a) = 1\} \tag{6}$$

$$= \bigcup_{\sum_{s,a} b_{sa}^p \leq \beta^p,} \{P \mid \|P_{sa} - \hat{P}_{sa}\|_p^p \leq b_{sa}^p, \sum_{s'} P(s'|s,a) = 1\} \tag{7}$$

$$= \bigcup_{b \in \mathcal{B}} \mathcal{U}_p^{sa}(b). \tag{8}$$

$\square$

## C Additional Results: Robust Policy Improvement

In the previous section, we identified that the worst-case kernel can be expressed as a rank-one perturbation of the nominal kernel. Leveraging this structure, we developed a method to efficiently evaluate the robust policy. This method also computes the perturbation ($\beta k^\top$) and, consequently, the worst-case kernel.

Using the computed worst kernel, we can directly evaluate the gradient with respect to the policy. This enables policy improvement through gradient ascent, as detailed in [33]:

$$\pi_{n+1} = \text{proj}\left[\pi_n + \eta_k \nabla_\pi J_{P_n}^\pi \big|_{\pi=\pi_n}\right], \tag{9}$$

where $P_n$ is the worst-case kernel estimate for the policy $\pi_k$. This method guarantees global convergence with an iteration complexity of $O(\epsilon^{-4})$ [33].

Alternatively, the policy gradient can be derived for the approximate perturbation, as shown in the result below.

**Policy Gradient Theorem**  Once the worst kernel for a policy is computed using Algorithm 1, the policy gradient can be used to update the policy. Alternatively, the following policy gradient theorem provides a direct way to compute the gradient:

**Lemma C.1** (Approximate Policy Gradient Theorem). *Given a transition kernel $P = \hat{P} - \beta k^\top$, the return is expressed as:*

$$J_P^\pi := J_0^\pi - \gamma \frac{J_\beta^\pi \langle k, v_R^\pi \rangle}{1 + \gamma \langle k, v_\beta^\pi \rangle},$$

*and the gradient is given by:*

$$\nabla_\pi J_P^\pi = d^\pi \circ Q_R^\pi - \gamma \frac{k^\top v_R^\pi}{1 + \gamma k^\top v_\beta^\pi} d^\pi \circ Q_\beta^\pi - \gamma \frac{J_\beta^\pi (k^\top D^\pi)}{1 + \gamma k^\top v_\beta^\pi} \circ Q_R^\pi + \gamma^2 \frac{J_\beta^\pi (k^\top v^\pi)(k^\top D^\pi)}{(1 + \gamma k^\top v_\beta^\pi)^2} \circ Q_\beta^\pi.$$

*Proof.* The expression for the return follows directly from the inverse matrix theorem, as shown in [18]. The gradient is then derived using the policy gradient theorem [29] in the format used in [19].

$$\nabla_\pi J_P^\pi = d^\pi \circ Q_R^\pi - \gamma \frac{k^\top D^\pi R^\pi}{1 + \gamma k^\top D^\pi \beta^\pi} d_\mu^\pi \circ Q_\beta^\pi - \gamma \frac{\mu^\top D \beta^\pi}{1 + \gamma k^\top D^\pi \beta^\pi} d_k^\pi \circ Q_R^\pi$$

$$+ \gamma^2 \frac{\mu^\top D \beta^\pi k^\top D^\pi R^\pi}{(1 + \gamma k^\top D^\pi \beta^\pi)^2} d_k^\pi \circ Q_\beta^\pi,$$

$$= d^\pi \circ Q_R^\pi - \gamma \frac{k^\top v_R^\pi}{1 + \gamma k^\top v_\beta^\pi} d^\pi \circ Q_\beta^\pi - \gamma \frac{J_\beta^\pi (k^\top D^\pi)}{1 + \gamma k^\top v_\beta^\pi} \circ Q_R^\pi + \gamma^2 \frac{J_\beta^\pi (k^\top v^\pi)(k^\top D^\pi)}{(1 + \gamma k^\top v_\beta^\pi)^2} \circ Q_\beta^\pi.$$

$\square$

The main advantage of this policy gradient formulation is that terms like $J_\beta^\pi, v_\beta^\pi, Q_\beta^\pi$, along with the nominal terms $J_R^\pi, v_R^\pi, Q_R^\pi$, can be efficiently computed using Bellman operators and bootstrapping techniques.

**Interpretation of Gradient Terms** The approximate policy gradient reveals the interplay of various components in robust MDPs:

- The first term, $d^\pi \circ Q_R^\pi$, represents the nominal policy gradient, emphasizing actions with high rewards.
- The second term, $\gamma \frac{k^\top v_R^\pi}{1 + \gamma k^\top v_\beta^\pi} d^\pi \circ Q_\beta^\pi$, discourages policies that place significant weight on high-uncertainty Q-values, scaled by the vulnerability to adversarial actions.
- The last two terms, while more complex to interpret, further reflect the intricate dynamics of robust MDPs.

**Robust Policy Gradient Algorithm** The robust policy gradient algorithm (Algorithm 3) converges to an $\epsilon$-optimal policy within $O(\epsilon^{-8})$ iterations.

**Theorem C.2.** *The robust policy gradient method from [33] achieves global convergence within $O(\epsilon^{-4})$ iterations for the policy gradient step. Algorithm 1 computes the worst-case kernel in $O(n)$ iterations at step $n$. The total iteration complexity for global optimality is $O(\epsilon^{-8})$.*

Algorithm 3 employs a double-loop structure: the inner loop (Algorithm 1) computes the worst-case kernel for a fixed policy, while the outer loop updates the policy using the derived gradient. An actor-critic style alternative, where the kernel and policy are updated simultaneously, is left for future work.

---

**Algorithm 3** Robust Policy Gradient Algorithm

---

1: **while** not converged: $n = n + 1$ **do**
2:    Compute the worst-case kernel $P = \hat{P} - \beta k^\top$ for policy $\pi$ using Algorithm 1 with tolerance $\epsilon = \gamma^n$.
3:    Compute the policy gradient $G$ using Lemma C.1.
4:    Update policy: $\pi \leftarrow \text{proj}\left[\pi + \alpha_n G\right]$.
5: **end while**

---

**Extension to KL Entropy Uncertainty Sets.** For the KL uncertainty case, the worst kernel is given by $P^\pi_{\mathcal{U}^{sa}_{KL}} = (I - \gamma \hat{P}^\pi A^\pi)^{-1}$ where $A^\pi$ is a diagonal matrix [31]. If we can invert this matrix, then its possible to build upon it. We leave this for future work.

# D  Helper Results

**Proposition D.1** (Sherman–Morrison Formula [4].)**.** *If $A \in \mathbb{R}^{n \times n}$ invertible matrix, and $u, v \in \mathbb{R}^n$, then the matrix $A + uv^T$ is invertible if and only if $1 + v^T A^{-1} u \neq 0$:*

$$(A + uv^T)^{-1} = A^{-1} - \frac{A^{-1} u v^T A^{-1}}{1 + v^T A^{-1} u}.$$

**Proposition D.2.**

$$\sigma_q(v) := \min_{w \in \mathbb{R}} \|v - w\mathbf{1}\|_q, = \min_{\|k\|_p \leq 1, \mathbf{1}^T k = 0} k^T v$$

*Proof.* Follows directly from Lemma J.1 of [20]. $\qquad \square$

**Proposition D.3.** *For any vector $\|x\| = 1$, we have*

$$\max\{\|Proj_{\mathbb{R}^n_+}(x)\|, \|Proj_{\mathbb{R}^n_+}(-x)\|\} \geq \frac{1}{\sqrt{2}},$$

*where $\mathbb{R}^n_+$ is positive quadrant.*

*Proof.* For any vector $\|x\| = 1$, we have

$$\|x_+\|^2 + \|x_-\|^2 = \|x\|^2 = 1.$$

And $Proj_{\mathbb{R}^n_+}(x) = x_+$ and $Proj_{\mathbb{R}^n_+}(-x) = x_-$, the rest follows. $\qquad \square$

**Proposition D.4.** *For $\|k\|_p$ and $k^T \mathbf{1} = 0$, we have*

$$1 + \gamma k^T (I - \gamma P^\pi)^{-1} b^\pi \geq 0,$$

*for all $\pi$, $\|b\|_p \leq \beta, b \succeq 0$.*

*Proof.* This is true from the Sherman–Morrison formula as $J^\pi_{\hat{P} - bk^T}$ is finite, hence the denominator must be strictly greater than zero. $\qquad \square$

# E  Dual Formulation

**Lemma E.1** (Sa-rectangular Duality)**.** *For the sa-rectangular uncertainty set $\mathcal{U} = \mathcal{U}^{sa}_p(\beta)$ with radius vector $\beta \in \mathbb{R}^{\mathcal{S} \times \mathcal{A}}$, the robust return can be written as the following optimization problem,*

$$J^\pi_{\mathcal{U}} = J^\pi - \gamma \max_{\|k\|_p = 1, \mathbf{1}^T k = 0} \frac{\mu^T D^\pi \beta^\pi k^T D^\pi R^\pi}{1 + \gamma k^T D^\pi \beta^\pi},$$

*where $\beta^\pi_s = \sum_a \pi(a|s)\beta_{sa}$.*

*Proof.* From [18], we know that the worst kernel $P^\pi_{\mathcal{U}^{sa}_p(\beta)}$ for the uncertainty set $\mathcal{U}^{sa}_p(\beta)$ is a rank one-perturbation of $P$. In other words,

$$P^\pi_{\mathcal{U}^{sa}_p(\beta)} = P + \beta k^T$$

for some $k \in \mathbb{R}^{\mathcal{S}}$ satisfying $\|k\|_p = 1$ and $1^T k = 0$. This implies that it is enough to look for rank-one perturbations of the nominal kernel $\hat{P}$ in order to find the robust return. That is,

$$
\begin{aligned}
J^\pi_{\mathcal{U}^{sa}_p(\beta)} &= \min_{P \in \mathcal{U}^{sa}_p(\beta)} J^\pi_P \\
&= \min_{P = \hat{P} + \beta k^T, \|k\|_p = 1, 1^T k = 0} J^\pi_P, \qquad \text{(looking only at rank one perturbations)} \\
&= \min_{P = \hat{P} + \beta k^T, \|k\|_p = 1, 1^T k = 0} \mu^T D^\pi_P R^\pi \\
&= \min_{P = \hat{P} + \beta k^T, \|k\|_p = 1, 1^T k = 0} \mu^T (I - \gamma P^\pi)^{-1} R^\pi \\
&= \min_{\|k\|_p = 1, 1^T k = 0} \mu^T \left( I - \gamma (P^\pi + \beta^\pi k^T) \right)^{-1} R^\pi \\
&= J^\pi - \gamma \max_{\|k\|_p = 1, 1^T k = 0} \frac{\mu^T D^\pi \beta^\pi k^T D^\pi R^\pi}{1 + \gamma k^T D^\pi \beta^\pi}.
\end{aligned}
$$

$\square$

**Lemma E.2** (S-rectangular Duality). *For $\mathcal{U} = \mathcal{U}^s_p$, the robust return can be written as the following optimization problem,*

$$
J^\pi_{\mathcal{U}} = J^\pi - \gamma \min_{\|\beta\|_p \le \epsilon, \|k\|_p \le 1, \langle 1, k \rangle = 0} \frac{\langle d^\pi, \beta^\pi \rangle \langle k, v^\pi \rangle}{1 + \gamma k^\top D^\pi \beta^\pi},
$$

*where $D^\pi = (I - \gamma P^\pi)^{-1}$, $d^\pi = \mu^T D^\pi$ and $v^\pi = D^\pi R^\pi$.*

*Proof.*

$$
\begin{aligned}
J^\pi_{\mathcal{U}^s_p(\beta)} &= \min_{\|P_s - (P)_s\|^p_p = \beta^p_s, 1^T P_{sa} = 1} J^\pi_P \\
&= \min_{\sum_a \beta^p_{sa} \le \beta^p_s} \min_{\|P_{sa} - (P)_{sa}\|_p = \beta_{sa}, 1^T P_{sa} = 1} J^\pi_P \\
&= \min_{\sum_a \beta^p_{sa} \le \beta^p_s} J^\pi_{\mathcal{U}^{sa}_p(\beta)} \\
&= \min_{\sum_a \beta^p_{sa} \le \beta^p_s} \left[ J^\pi - \gamma \max_{\|k\|_p = 1, 1^T k = 0} \frac{\mu^T D^\pi \beta^\pi k^T D^\pi R^\pi}{1 + \gamma k^T D^\pi \beta^\pi} \right] \\
&= J^\pi - \gamma \max_{\sum_a \beta^p_{sa} \le \beta^p_s, \|k\|_p = 1, 1^T k = 0} \frac{\mu^T D^\pi \beta^\pi k^T D^\pi R^\pi}{1 + \gamma k^T D^\pi \beta^\pi}.
\end{aligned}
$$

$\square$

The above result formulates the robust return in terms of nominal values only for the first time. This implies the robust objective can be rewritten in the dual form as :

$$
J^*_{\mathcal{U}^s_p} = \max_{D \in \mathcal{D}} \min_{k \in \mathcal{K}, b \in \mathcal{B}} \left[ \mu^T D R^\pi - \gamma \mu^T D b^\pi \frac{k^T D R^\pi}{1 + \gamma k^T D b^\pi} \right]
$$

where $\mathcal{D} = \{(I - \gamma P^\pi_0)^{-1} \mid \pi \in \Pi\}$, $\mathcal{K} = \{k \in \mathbb{R}^{\mathcal{S}} \mid \|k\|_p = 1, 1^T k = 0\}$, and $\mathcal{B} = \{b \in R^{\mathcal{S} \times \mathcal{A}} \mid \|b_s\|_p \le \beta_s\}$.

Comparing the penalty term from the previous results in [20, 18], the dual formulation can be written as

$$
J^*_{\mathcal{U}^s_p} = \max_{D \in \mathcal{D}} \min_{k \in \mathcal{K}} \left[ \mu^T D R^\pi - \gamma \mu^T D \beta^\pi \frac{k^T D R^\pi}{1 + \gamma k^T D \beta^\pi} \right]
$$

where $\beta^\pi_s = \|\pi_s\|_q \beta_s$.

Surprisingly, the optimization here looks as if it is optimized for the same value of $\beta^\pi_s = \max_{\sum_a \beta^p_{sa} \le \beta^p_s} \sum_a \pi(a|s) \beta_{sa} = \beta_s \|\pi_s\|_q$ for all values of feasible $k$. This suggest that the adversary payoff is maximized by maximizing the expected uncertainty in the trajectories.

**Lemma E.3** (Non-rectangular Duality). *For $\mathcal{U} = \mathcal{U}_p$, the robust return can be written as the following optimization problem*

$$J_{\mathcal{U}}^{\pi} = J^{\pi} - \gamma \min_{\|\beta\|_p \leq \epsilon, \|k\|_p \leq 1, \langle 1, k \rangle = 0} \frac{\langle d^{\pi}, \beta^{\pi} \rangle \langle k, v_R^{\pi} \rangle}{1 + \gamma \langle k, v_{\beta}^{\pi} \rangle},$$

*where $D^{\pi} = (I - \gamma P^{\pi})^{-1}$, $d^{\pi} = \mu^T D^{\pi}$ and $v^{\pi} = D^{\pi} R^{\pi}$.*

*Proof.* Now,

$$J_{\mathcal{U}_p(\epsilon)}^{\pi} = \min_{\|P - P\|_p^p = \epsilon^p, 1^T P_{sa} = 1} J_P^{\pi}$$

$$= \min_{\|\beta\|_p^p \leq \epsilon^p} \min_{\|P_{sa} - (P)_{sa}\|_p = \beta_{sa}, 1^T P_{sa} = 1} J_P^{\pi}$$

$$= \min_{\|\beta\|_p^p \leq \epsilon^p} J_{\mathcal{U}_p^{sa}(\beta)}^{\pi}$$

$$= \min_{\|\beta\|_p \leq \epsilon} \left[ J^{\pi} - \gamma \max_{\|k\|_p = 1, 1^T k = 0} \frac{\mu^T D^{\pi} \beta^{\pi} k^T D^{\pi} R^{\pi}}{1 + \gamma k^T D^{\pi} \beta^{\pi}} \right]$$

$$= J^{\pi} - \gamma \max_{\|\beta\|_p \leq \epsilon, \|k\|_p = 1, 1^T k = 0} \frac{\mu^T D^{\pi} \beta^{\pi} k^T D^{\pi} R^{\pi}}{1 + \gamma k^T D^{\pi} \beta^{\pi}}.$$

$\square$

The above result formulates the robust return in terms of nominal values only, for the first time. Comparing with the existing result, we get a very interesting relation:

$$\sigma_q(v_{\mathcal{U}}^{\pi}) = \max_{\|k\|_p = 1, 1^T k = 0} \frac{k^T v_R^{\pi}}{1 + \gamma k^T v_{\beta}^{\pi}}, \tag{10}$$

where $v_x^{\pi} = (I - \gamma P^{\pi})^{-1} x^{\pi}$.

The LHS is a robust quantity (variance of the robust return) which is express in the terms of purely nominal quantities. This is the simplest of all such relations. We believe that the above relation can help in theoretical derivations and experiment design but not exactly sure how yet.

### E.1 Intuition on the Adversary

**sa-rectangular case.** We know that the $\sigma(v_{\mathcal{U}}^{\pi})$ represents the penalty for robustness, expressed as:

$$J_{\mathcal{U}}^{\pi} = J^{\pi} - \gamma \langle d^{\pi}, \beta^{\pi} \rangle \sigma_q(v_{\mathcal{U}}^{\pi}).$$

Understanding how $\sigma(v_{\mathcal{U}}^{\pi})$ arises provides insight into the behavior of the adversary as described in (10). Furthermore, if $P = \hat{P} - \beta k^T$, then:

$$J_P^{\pi} = J^{\pi} - \langle d^{\pi}, \beta^{\pi} \rangle \frac{k^T v_R^{\pi}}{1 + \gamma k^T v_{\beta}^{\pi}}.$$

Here, $k$ represents the direction in which the adversary discourages perturbations in the kernel. The optimal direction $k$ chosen by the adversary maximizes the objective in (10).

**s-rectangular uncertainty sets.** Now, we turn our attention to the coupled uncertainty case.

**Lemma E.4.** *For $\mathcal{U} = \mathcal{U}_p^s$, the robust return can be formulated as the following optimization problem:*

$$J_{\mathcal{U}}^{\pi} = J^{\pi} - \gamma \min_{\|\beta\|_p \leq \epsilon, \|k\|_p \leq 1, \langle 1, k \rangle = 0} \frac{\langle d^{\pi}, \beta^{\pi} \rangle \langle k, v^{\pi} \rangle}{1 + \gamma k^\top D^{\pi} \beta^{\pi}},$$

*where $D^{\pi} = (I - \gamma P^{\pi})^{-1}$, $d^{\pi} = \mu^T D^{\pi}$, and $v^{\pi} = D^{\pi} R^{\pi}$.*

*Proof.* The proof follows similarly to the `sa`-rectangular case and is detailed in the appendix. The key additional step involves decomposing the `s`-rectangular uncertainty set $\mathcal{U}_p^s$ into a union of `sa`-rectangular uncertainty sets $\mathcal{U}_p^{sa}$. $\qquad\square$

By comparing the penalty term from previous results in [20, 18], we obtain:

$$\sum_s d^\pi(s)\|\pi_s\|_q \sigma_q(v_{\mathcal{U}}^\pi) = \max_{\sum_a \beta_{sa}^p \le \beta_s^p, \|k\|_p = 1, 1^T k = 0} \frac{(d^\pi \beta^\pi)(k^T v^\pi)}{1 + \gamma k^T D^\pi \beta^\pi}.$$

This relation is interesting as it connects the robust term on the left-hand side (LHS) with the non-robust terms on the right-hand side (RHS).

Interestingly, the optimization here suggests that the adversary maximizes the expected uncertainty in trajectories, as the same value of $\beta_s^\pi = \max_{\sum_a \beta_{sa}^p \le \beta_s^p} \sum_a \pi(a|s)\beta_{sa} = \beta_s\|\pi_s\|_q$ appears for all feasible $k$.

## F   Robust Policy Evaluation

**Proposition F.1.** *For* $\lambda^* = \max_{x \in C} \frac{g(x)}{h(x)}$ , $F(\lambda) := \max_{x \in C} \Big( g(x) - \lambda h(x) \Big)$ , *we have* $F(\lambda^*) = 0$ *and* $f(\lambda) \ge 0 \iff \lambda^* \ge \lambda$.

*Proof.*     • If $F(\lambda) \ge 0$ then

$$\exists x \quad s.t. \quad g(x) - \lambda h(x) \ge 0$$
$$\implies \exists x \quad s.t. \quad \frac{g(x)}{h(x)} \ge \lambda, \qquad (\text{as } h(x) > 0 \text{ for all } x)$$
$$\implies \max_{x \in C} \frac{g(x)}{h(x)} \ge \lambda.$$

• If $F(\lambda) \le 0$ then

$$g(x) - \lambda h(x) \le 0, \qquad \forall x \in C$$
$$\implies \frac{g(x)}{h(x)} \le \lambda, \qquad \forall x \in C, \qquad (\text{as } h(x) > 0)$$
$$\implies \max_{x \in C} \frac{g(x)}{h(x)} \le \lambda$$

• If $F(\lambda) = 0$ then $\lambda = \max_{x \in C} \frac{g(x)}{h(x)}$ implied from the above two items.

$\qquad\square$

**Lemma F.2.** *The robust return can be expressed as*

$$J_{\mathcal{U}_p}^\pi = J^\pi - \lambda^*,$$

*where the penalty* $\lambda^*$ *is a fixed point of* $F(\lambda)$. *Furthermore,* $\lambda^*$ *can be found via binary search as* $F(\lambda) > \lambda$ *if and only if* $\lambda > \lambda^*$, *where* $F(\lambda) = \max_{b \in \mathcal{B}} \|E^\pi b\|_q$, $E^\pi = \gamma \Big( I - \frac{\mathbf{1}\mathbf{1}^\top}{S} \Big) \Big[ D^\pi R^\pi \mu^\top D^\pi - \lambda D^\pi \Big] H^\pi$, *and* $H^\pi R := R^\pi$.

*Proof.* We want to evaluate the following

$$\lambda^* := \max_{b \in \mathcal{B}, k \in \mathcal{K}} \gamma \frac{k^T D^\pi R^\pi \mu^T D^\pi b^\pi}{1 + \gamma k^T D^\pi b^\pi}.$$

This is of the form $\max_x \frac{f(x)}{g(x)}$ . Then according to Proposition F.1, we have $f(\lambda^*) = 0$ and $f(\lambda) > 0$ if and only if $\lambda^* > \lambda$, where

$$f(\lambda) := \max_{b \in \mathcal{B}, k \in \mathcal{K}} \left[ \gamma k^T A^\pi b^\pi - \lambda(1 + \gamma k^T D^\pi b^\pi) \right]$$

$$= \max_{b \in \mathcal{B}, k \in \mathcal{K}} k^\top C^\pi b - \lambda,$$

$$= \max_{b \in \mathcal{B}, \|k\|_p \le 1} k^\top \left( I - \frac{\mathbf{1}\mathbf{1}^T}{S} \right) C^\pi b - \lambda, \quad \text{(from Proposition G.6)}$$

$$= \max_{b \in \mathcal{B}} \| \left( I - \frac{\mathbf{1}\mathbf{1}^T}{S} \right) C^\pi b \|_q - \lambda, \quad \text{(Holder's inequality)}$$

where $A^\pi = D^\pi R^\pi \mu^T D^\pi, C^\pi := \gamma \left( A^\pi - \lambda D^\pi \right) H^\pi$.

$\square$

# G  Evaluation of $\max_{x,y} xEy$

In Algorithm 1 , we need to solve

$$z_p := \max_{\|b\|_p \le 1, \|k\|_p \le 1, b \succeq 0, <k,1>=0} k^\top E b$$

as sub-routine which is a bi-linear optimization. And bilinear optimization is NP-Hard in general. But we have a special form, which might be solvable for many cases, as illustrate below. As we see later, for $L_1$ bounded uncertainty set, we provided provably efficient robust-policy evaluation method that provides the solution in the close form. For $p = 2$, we provided heuristic algorithm that works well in practice and also generalizes well beyond theoretically applicable uncertainty radius $\beta$. However, we believe it is possible provable solve this sub-problem via standard methods as second-order cone optimization or quadratic programming, but we are not aware of the exact positive nor negative result about this case. For $p = \infty$, non-rectangular uncertainty set degenerates to sa-rectangular one, which is already efficiently solvable. And for other $p$, we are not sure if the efficient solution exists.

## G.1  For $p = 1$

For uncertainty case $\mathcal{U}_1$ bounded by $L_1$-ball. We have the close-form solution to our sub-problem.

**Lemma G.1.**
$$\max_{\|b\|_1 \le 1, \|k\|_1 \le 1, b \succeq 0, <k,1>=0} k^\top E b = \max_i \|E(i, \cdot)\|_{sp},$$

where $\|x\|_{sp} = \frac{\max_{x_i} - \min_i x_i}{2}$ is span-norm.

*Proof.*

$$\max_{\|b\|_1 \le 1, \|k\|_1 \le 1, b \succeq 0, <k,1>=0} k^\top E b = \max_{\|b\|_1 \le 1, b \succeq 0} \max_{\|k\|_1 \le 1, <k,1>=0} k^\top E b \tag{11}$$

$$= \max_{\|b\|_1 \le 1, b \succeq 0} \|E b\|_{sp}, \quad \text{(from Preposition G.2 )} \tag{12}$$

$$= \max_i \|E(\cdot, i)\|_{sp}, \quad \text{(from Preposition G.3).} \tag{13}$$

Notably, the optimizers $k^*, b^*$ are given as

$$b^*(i) = \mathbf{1}(i = i^*),$$

where $i^* = \arg\max_i \|E(\cdot, i)\|_{sp}$, and

$$k^*(i_{max}) = -k^*(i_{min}) = \frac{1}{2},$$

where $i_{max} = \arg\max_i (Eb^*)(i)$ and $i_{min} = \arg\min_i (Eb^*)(i)$.

$\square$

**Proposition G.2.**

$$\max_{\|k\|_1=1, \langle \mathbb{1}, k \rangle=0} \langle k, x \rangle = \|x\|_{sp},$$

*where* $\|x\|_{sp} = \frac{\max_i x_i - \min_i x_i}{2}$.

*Proof.* Let $x \in \mathbb{R}^n$, then we have

$$\max_{\|k\|_1=1, \langle \mathbb{1}, k \rangle=0} \sum_i k_i x_i = \max_{\|k\|_1=1, \langle \mathbb{1}, k \rangle=0} \left[ \sum_{i:k_i>0} k_i x_i + \sum_{i:k_i<0} k_i x_i \right] \tag{14}$$

$$= \max_{\sum_{i:k_i>0} k_i = -\sum_{i:k_i<0} k_i = \frac{1}{2}} \left[ \sum_{i:k_i>0} k_i x_i + \sum_{i:k_i<0} k_i x_i \right] \tag{15}$$

$$= \frac{\max_i x_i - \min_i x_i}{2}. \tag{16}$$

$\square$

**Proposition G.3.**

$$\max_{\|b\|_1=1, b \succeq 0} \|Eb\|_{sp} = \max_i \|E(\cdot, i)\|_{sp},$$

*where* $\|x\|_{sp} = \frac{\max_i x_i - \min_i x_i}{2}$.

*Proof.*

$$\max_{\|b\|_1=1, b \succeq 0} \|Eb\|_{sp} = \max_{\|b\|_1=1, b \succeq 0} \left\| \sum_i E(\cdot, i) b_i \right\|_{sp}, \qquad \text{(matrix multiplication)} \tag{17}$$

$$\leq \max_{\|b\|_1=1, b \succeq 0} \sum_i b_i \|E(\cdot, i)\|_{sp}, \qquad (\|\cdot\|_{sp} \text{ satisfies triangle inequality}) \tag{18}$$

$$= \max_i \|E(\cdot, i)\|_{sp}, \qquad \text{(maximization over } b). \tag{19}$$

Further, $\max_i \|E(\cdot, i)\|_{sp}$ is achievable by taking $b$ to one-hot vector at index $i$ that maximizes $\max_i \|E(\cdot, i)\|_{sp}$. Hence, we get the desired result. $\square$

**Proposition G.4.** *For $p = 1$, $F$ can be evaluated in closed form as*

$$F(\lambda) = \gamma \max_{s,a} \pi(a|s) \|d^\pi(s) v^\pi - \lambda D^\pi(\cdot, s)\|_{sp},$$

*where* $\|x\|_{sp} = \frac{\max_i x_i - \min_i x}{2}$.

*Proof.* From Lemma G.1, we have $F(\lambda) = \max_{s,a} \|E_\lambda^\pi(\cdot, (s, a))\|_{sp}$. And from definition, we have $E_\lambda^\pi := \gamma \left[ D^\pi R^\pi \mu^\top D^\pi - \lambda D^\pi \right] H^\pi$. Further,

$$\left( D^\pi R^\pi \mu^\top D^\pi H^\pi \right) \left( s', (s, a) \right) = \left( v^\pi (d^\pi)^\top H^\pi \right) \left( s', (s, a) \right) \tag{20}$$

$$= \left( v^\pi (d^\pi(\cdot, \cdot))^\top \right) \left( s', (s, a) \right) \tag{21}$$

$$= v^\pi(s') d^\pi(s) \pi(a|s). \tag{22}$$

Further,

$$\left( D^\pi H^\pi \right) \left( s', (s, a) \right) = \sum_x d_{s'}^\pi(x) H^\pi(x, (s, a)) = d_{s'}^\pi(s) \pi(a|s). \tag{23}$$

Finally, using $\|x\|_{sp} = \max_{i,j} \frac{x_i - x_j}{2}$, we get the desired result.

$\square$

## G.2  For $p = 2$

For $p = 2$, we proposed a heuristic spectral Algorithm 2, to solve the above sub-problem. And as the Figure 6, Figure 7 and Table 3 shows the superiority of its performance over numerical methods (scipy). Further, we showed in the previous response that this method yields much better performance than random search even when uncertainty radius $\beta$ is big-enough.

For other $p \neq 1, 2$, we are not sure if the above sub-problem is efficiently sovable or not. As a similar problem without positivity and mean-zero constraints,

$$A_{p \to q} := \max_{\|b\|_p \leq 1, \|k\|_p \leq 1} b^\top A k = \max_{\|b\|_p \leq 1} \|b^\top A\|_q,$$

is very well studied problems, and several negative results are known in the literature for $p \neq 1, 2, \infty$ [13].

**Proposition G.5.** *[Orthogonality Equivalence]Let* $\mathcal{K} = \{k \mid \|k\|_2 \leq 1, 1^\top k = 0\}$, *and* $\mathcal{W} = \{k^T(I - \frac{11^T}{S}) \mid \|k\|_2 \leq 1\}$ . *Then we have,*

$$\mathcal{K} = \mathcal{W}.$$

*Proof.* Now let $k \in \mathcal{K}$, then $k^T(I - \frac{11^T}{S}) = k^\top \in \mathcal{W}$. Now the other direction, let $k \in \mathcal{W}$, then $\langle k^T(I - \frac{11^T}{S}), 1 \rangle = 0$ by construction and $\|k^T(I - \frac{11^T}{S})\|_p \leq \|k\|_2 \leq 1$, this implies $k^T(I - \frac{11^T}{S}) \in \mathcal{K}$. $\qquad\square$

The above result implies that

$$\max_{\|b\|_2 \leq \beta, \|k\|_p \leq 1, 1^T k = 0} k^T A b = \max_{\|b\|_2 \leq \beta, k \in \mathcal{K}} k^T A b$$

$$= \max_{\|b\|_2 \leq \beta, k \in \mathcal{W}} k^T A b, \qquad \text{(as } \mathcal{K} = \mathcal{W} \text{ from above Proposition G.5)}$$

$$= \max_{\|b\|_2 \leq \beta, \|k\|_2 = 1} k^\top (I - \frac{11^T}{S}) A b, \qquad \text{(def. of } \mathcal{W}).$$

Further, we have equivalence of optimizers

$$\operatorname*{arg\,max}_{\|k\|_2 \leq 1, 1^T k = 0, \|b\|_2 \leq \beta} k^T A b = \left\{ (b^*, (I - \frac{11^T}{S}) k^*) \mid (b^*, k^*) \in \operatorname*{arg\,max}_{\|k\|_2 = 1, \|b\|_2 \leq \beta} k^\top (I - \frac{11^T}{S}) A b \right\}.$$

**Proposition G.6.** *The solving of*

$$\max_{\|k\|_2 \leq 1, 1^T k = 0, \|b\|_2 \leq \beta} k^T A b, \quad \text{is equivalent to} \quad \max_{\|k\|_2 = 1, \|b\|_2 \leq \beta} k^\top (I - \frac{11^T}{S}) A b.$$

*Proof.* Directly follows from the proposition above. $\qquad\square$

## G.3  Eigenvalue Approach (Spectral Methods)

This section focus on deriving a spectral method for solving the optimization problem:

$$\max_{\|x\|_2 \leq 1, \, x \geq 0} \|Ax\|_2,$$

where $A \in \mathbb{R}^{n \times n}$. Compute $A^\top A$. We perform eigenvalue decomposition of $A^\top A$:

$$A^\top A = V \Lambda V^\top,$$

where $\Lambda = \operatorname{diag}(\lambda_1, \lambda_2, \ldots, \lambda_n)$ (eigenvalues) and $V = [v_1, v_2, \ldots, v_n]$ (eigenvectors). Further, WLOG

$$\lambda_1 \geq \lambda, \cdots, \qquad , \text{and} \quad \|v+_i\| \geq \|v-_i\| \qquad \forall i, \qquad u_i := \frac{v_i^+}{\|v_i^+\|}$$

where $v_i^+ = \max(v_i, 0)$, $\quad v_i^- = -\min(v_i, 0)$ denotes positive and negative parts respectively.

- Zero Order Solution:
$$f_0 = \|Au_1\|.$$

- First order solution:
$$f_1 = \max_i \|Au_i\|.$$

- Second order solution:
$$f_2 = \max_{i,j} \max_{t \in [0,1]} \|A \frac{(tv_i + (1-t)v_j)^+}{\|(tv_i + (1-t)v_j)^+\|}\|.$$

- Third order solution:
$$f_3 = \max_{i,j,k} \max_{r,s,t, \in [0,1], r+s+t=1} \|A \frac{(rv_i + sv_j + tv_k)^+}{\|(rv_i + sv_j + tv_k)^+\|}\|.$$

Upper bounds on $\max_{\|x\|_2 \le 1, x \succeq 0} \|Ax\|_2$:

- Zero order upper bound: $\lambda_1$
- First order upper bound: $\sqrt{\sum_i \lambda_i c_i}$, where $c_i =$
$$\begin{cases} \langle v_i, u_i \rangle^2, & \text{if } \sum_{j=1}^{i} \langle v_i, u_i \rangle^2 \le 1 \\ 1 - \sum_{j=1}^{i-1} \langle v_i, u_i \rangle^2, & \text{if } \sum_{j=1}^{i} \langle v_i, u_i \rangle^2 \ge 1, \sum_{j=1}^{i-1} \langle v_i, u_i \rangle^2 \le 1 \, . \\ 0 & \text{otherwise} \end{cases}$$

**Lemma G.7** (Zero Order Approximation). *The highest projected eigenvector $u = \frac{v_1^+}{\|v_1^+\|}$ is at least a half-good solution, i.e.,*
$$\|Au\|_2^2 \ge \frac{\lambda_1}{2} \ge \frac{1}{2} \max_{\|x\|_2 \le 1, \, x \ge 0} \|Ax\|_2^2.$$

*Further, if $A$ is rank-one then it is exact, i.e.,*
$$\|Au\|_2 = \max_{\|x\|_2 \le 1, \, x \ge 0} \|Ax\|_2.$$

*Proof.* We have $\|v_1^+\| \ge \frac{1}{\sqrt{2}}$ from Proposition D.3. Let $u = \frac{(v_1)_+}{\|(v_1)_+\|} = \sum_i \sigma_i v_i$, where $\sigma_i = \langle u, v_i \rangle$, we have

$$\begin{aligned} u^T A^T A u &= (\sum_i \sigma_i v_i)(\sum_i \lambda_i v_i v_i^T)(\sum_i \sigma_i v_i) \\ &= \sum_i \lambda_i \sigma_i^2, \qquad \text{(as } v_i \text{ are orthogonal)} \\ &= \lambda_1 \sigma_1^2 + \sum_{i \ne 1} \lambda_i \sigma_i^2, \\ &\ge \lambda_1 \sigma_1^2 + \sum_{i \ne 1} \lambda_n \sigma_i^2, \qquad \text{(as } \lambda_2 \ge \lambda_3, \cdots) \\ &= \lambda_1 \sigma_1^2 + \lambda_n (1 - \sigma_1^2), \qquad \text{(as } \sum_i \sigma_i^2 = 1) \\ &\ge \frac{1}{2}(\lambda_1 + \lambda_n), \qquad \text{(as } \sigma_1 \ge \frac{1}{\sqrt{2}}). \end{aligned}$$

Rest follows.

$\square$

**Proposition G.8** (First Order is Better than the First).
$$\|Au_j\|_2^2 \ge \max_i \lambda_i \sigma_i^2 \ge \frac{\lambda_1}{2}$$

*where $j \in \arg\max_i \lambda_i \langle v_i, u_i \rangle$ and $\sigma_i = \langle v_i, u_i \rangle \ge \frac{1}{\sqrt{2}}$.*

*Proof.* Let $u_j = \frac{(v_j)_+}{\|(v_j)_+\|} = \sum_i \sigma_i^j v_i$, where $\sigma_i^j = \langle u_j, v_i \rangle$, we have

$$
\begin{aligned}
u_j^T A^T A u_j &= (\sum_i \sigma_i^j v_i)(\sum_i \lambda_i v_i v_i^T)(\sum_i \sigma_i^j v_i) \\
&= \sum_i \lambda_i (\sigma_i^j)^2, \qquad \text{(as } v_i \text{ are orthogonal)}, \\
&\geq \lambda_j (\sigma_j^j)^2, \\
&= \max_i \lambda_i (\sigma_i)^2, \qquad \text{(by definition of } j\text{)}.
\end{aligned}
$$

Rest follows.

$\square$

**Proposition G.9.** *Second order solution* $f_2 = \max_{i,j} \max_{t \in [0,1]} \|A \frac{(tv_i + (1-t)v_j)^+}{\|(tv_i + (1-t)v_j)^+\|}\|$ *is exactly equal to* $\max_{\|x\|_2 \leq 1, x \succeq 0} \|Ax\|_2$ *when* $A$ *is rank two.*

This approach is computationally efficient but may not always yield the exact solution, especially when multiple eigenvectors significantly contribute to the optimal $x$.

The intuition behind this approach is that the matrix $A^\top A$ can be decomposed into its eigenvalues and eigenvectors, representing the principal directions of the transformation applied by $A$. The eigenvector corresponding to the largest eigenvalue provides the direction of maximum scaling for $A$. However, since the solution is constrained to the nonnegative orthant ($x \geq 0$), we adjust the eigenvectors by only considering their positive parts. The method identifies an approximate solution $u_j$ by selecting and normalizing the positive part of the eigenvector that contributes the most to the objective function.

---

**Algorithm 4** Second Order Spectral Approximation for $\max_{\|x\|_2 \leq 1, x \geq 0} \|Ax\|_2$

---

1: Normalize the positive part:
$$
u_i = \frac{v_i^+}{\|v_i^+\|_2}.
$$

2: Compute scores for all eigenvectors:
$$
\text{Score}_i = \lambda_i \langle v_i, u_i \rangle.
$$

3: Select $j = \arg\max_i \text{Score}_i$.
4: **Output:** Approximate solution $u_j = v_j^+ / \|v_j^+\|_2$ and approximate maximum value $\|Au_j\|_2$.

---

**Notes**

- This approach is effective when the largest eigenvalue $s_1$ dominates the others. It approximates the solution by leveraging the spectral properties of $A^\top A$.

- The result might not be exact if multiple eigenvalues contribute significantly, as the approach considers only the contribution of individual eigenvectors.

## G.4 Experimental Verification

This section describes three different methods for solving the optimization problem:

$$
\max_{\|x\|_2 \leq 1, \, x \geq 0} \|Ax\|_2,
$$

where $A \in \mathbb{R}^{n \times n}$. The methods are compared in terms of their computational efficiency and the quality of their solutions.

### G.4.1 Brute Force Random Search

The brute force method randomly samples vectors $x \in \mathbb{R}^n$ from the nonnegative orthant, normalizes them to satisfy $\|x\|_2 = 1$, and evaluates $\|Ax\|_2$ for each sampled vector. The steps are as follows:

1. Generate $N$ random vectors $x_i \geq 0$, $i = 1, \ldots, N$.
2. Normalize each vector to unit norm: $x_i \leftarrow x_i/\|x_i\|_2$.
3. Compute $\|Ax_i\|_2$ for each vector and select the maximum value.

This method is simple to implement but computationally expensive, as it evaluates $A$ for a large number of randomly generated vectors. See figure 6

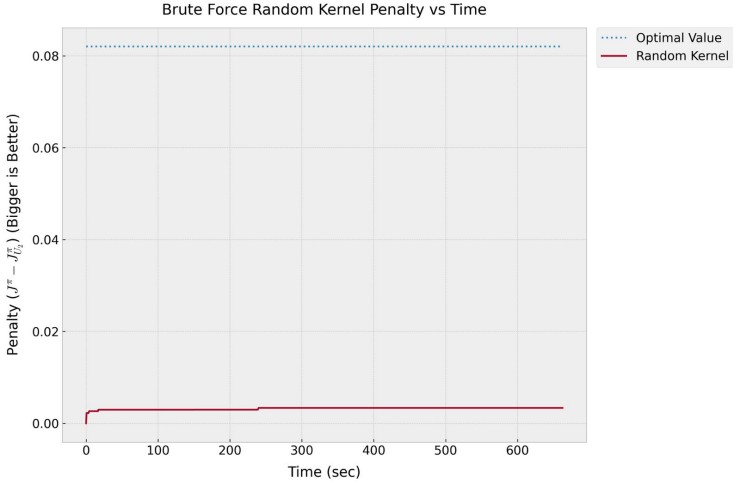

Figure 6: Random Kernel Guess takes exponentially long time to converge. While Algorithm 1 only took 0.14 sec to find the optimal value.

### G.4.2 Numerical Optimization (Scipy Minimize)

This approach uses numerical optimization to directly solve the problem:

$$\max_{\|x\|_2 \leq 1, \, x \geq 0} \|Ax\|_2.$$

The optimization problem is formulated as:

$$\min_x -\|Ax\|_2, \quad \text{subject to } \|x\|_2 \leq 1 \text{ and } x \geq 0.$$

Steps include:

1. Define the objective function as $-\|Ax\|_2$.
2. Impose constraints: $\|x\|_2 \leq 1$ and $x \geq 0$.
3. Solve the problem using `scipy.optimize.minimize`, with an initial guess $x_0$.

This method provides the exact solution but is computationally more expensive than the spectral method.

### G.5 Comparison Metrics

The three methods are compared based on:

- **Optimality:** The maximum value $\|Ax\|_2$ achieved by each method.
- **Time Efficiency:** The computational time required by each method.

### G.6 Results and Observations

The following plots compare the performance of the three methods:

- **Optimality Plot:** Shows that the maximum value obtained with scipy.minimize is slightly better than our spectral method, while random search performs poorly.
- **Time Efficiency Plot:** Illustrates the that scipy.minimize scales much poorly with the dimension, while our spectral method is way faster than both methods.

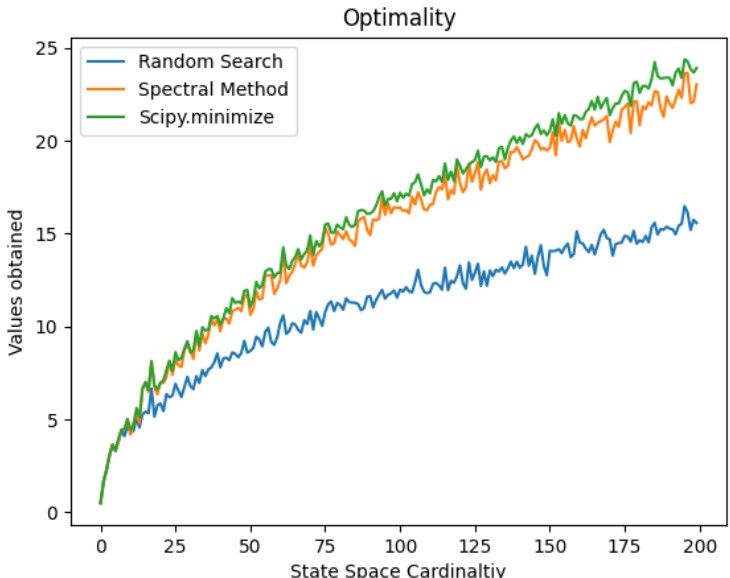

Figure 7: Comparison of optimality across methods.

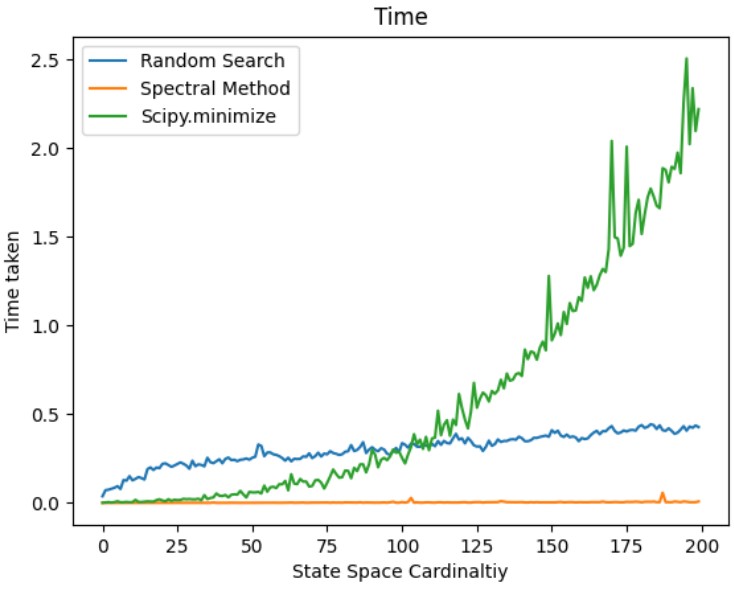

Figure 8: Comparison of computational time across methods

| | Optimal values attained | | | Time taken | | |
|---|---|---|---|---|---|---|
| $n$ | Random | Spectral | minimize | Random | Spectral | minimize |
| 10 | 4.10 | 4.45 | 4.46 | 0.12 | 0.0007 | 0.005 |
| 20 | 5.14 | 6.71 | 6.82 | 0.19 | 0.0003 | 0.01 |
| 50 | 9.23 | 11.59 | 11.93 | 0.25 | 0.0007 | 0.03 |
| 100 | 11.95 | 16.44 | 17.19 | 0.31 | 0.001 | 0.28 |
| 200 | 15.74 | 22.1 | 23.68 | 0.44 | 0.004 | 2.1 |
| 300 | 19.32 | 28.58 | 29.73 | 0.57 | 0.012 | 8.19 |
| 500 | 24.46 | 36.56 | 38.47 | 0.83 | 0.209 | 43.49 |
| 1000 | 33.91 | 51.64 | 54.25 | 1.38 | 0.171 | 313.6 |

Table 3: Attained Values and Time Taken.

### G.6.1 Parameters of Experiments

The experiments were conducted to evaluate the performance of three methods—brute force random search, eigenvalue heuristic, and numerical optimization—on solving the problem:

$$\max_{\|x\|_2 \leq 1, \, x \geq 0} \|Ax\|_2.$$

### State Space Cardinality and Random matrix Generation

- **State Space Cardinality** ($n$)**:** The dimension of the problem, denoted by $n$, represents the state space cardinality. In the experiments, $n$ varied from 1 to 300 to analyze the scalability of the methods.

- **Matrix Generation:** The matrix $A \in \mathbb{R}^{n \times n}$ was generated as a random matrix with entries sampled from a standard normal distribution:

$$A_{ij} \sim \mathcal{N}(0,1), \quad i, j = 1, \ldots, n.$$

  The same random seed (`seed = 42`) was used across all runs to ensure reproducibility.

- 10000 random vectors $x$ were generated for Brute Search Method.

**Process of matrix Evaluation**   The goal of the experiments is to maximize $\|Ax\|_2$ under the constraints $\|x\|_2 \leq 1$ and $x \geq 0$. The matrix $A$ is evaluated by:

1. Generating random vectors $x \in \mathbb{R}^n$ for the brute force method.
2. Computing the spectral decomposition of $A^\top A$ for the eigenvalue heuristic.
3. Defining and solving a constrained optimization problem for the numerical optimization method.

The results, including the optimal values and computational times, are recorded for each method.

**Evaluation Metrics**   The performance of the methods was assessed using the following metrics:

- **Optimality:** The maximum value $\|Ax\|_2$ obtained by each method.
- **Computational Efficiency:** The time taken by each method to compute the result.
- **Scalability:** The behavior of the methods as $n$ increases.

This systematic evaluation ensures a fair comparison of the three approaches across varying problem sizes.

**Hardware and Software Specifications**   The experiments were conducted on the following hardware and software setup:

- **Model Name:** MacBook Pro (2023 model).
- **Model Identifier:** Mac14,7.

- **Chip:** Apple M2 with 8 cores (4 performance and 4 efficiency cores).
- **Memory:** 16 GB Unified Memory.
- **Operating System:** macOS Ventura.
- **Programming Language:** Python 3.9.
- **Libraries Used:**
    - `numpy` for numerical computations.
    - `scipy` for numerical optimization.
    - `matplotlib` for generating plots.
    - `time` for recording computational times.

The experiments were designed to ensure reproducibility by fixing the random seed (`seed = 42`). Computational times and results are specific to the above hardware configuration and may vary on different systems.

# H  Convexity of $\mathcal{D}$

## H.1  MDP Configuration

We define an MDP with the following parameters:

- **State space size**: $S = 3$
- **Action space size**: $A = 2$
- **Discount factor**: $\gamma = 0.9$
- Random kernel $P$, random reward $R$, seed 42.
- Compute the set $\mathcal{D} = \{D^\pi H^\pi | \pi\}$ with 10 millions random policies $\pi$

## H.2  Dimensionality Reduction via PCA

Given the high-dimensional nature of the $D^\pi H^\pi$ representations, we apply **Principal Component Analysis (PCA)** to extract meaningful structure.

- We **retain the top 10 components** to capture the dominant variations in the dataset.
- The **explained variance ratio** is visualized to assess how much information each component retains.
- **2D projections** of the first few principal components are generated for visualization.

## H.3  Random Linear Projections

To further explore the **geometry of the occupancy measure set**, we apply **random linear projections** of the high-dimensional data:

- **2D Random Projections**: The data is projected onto **randomly chosen 2D subspaces**.

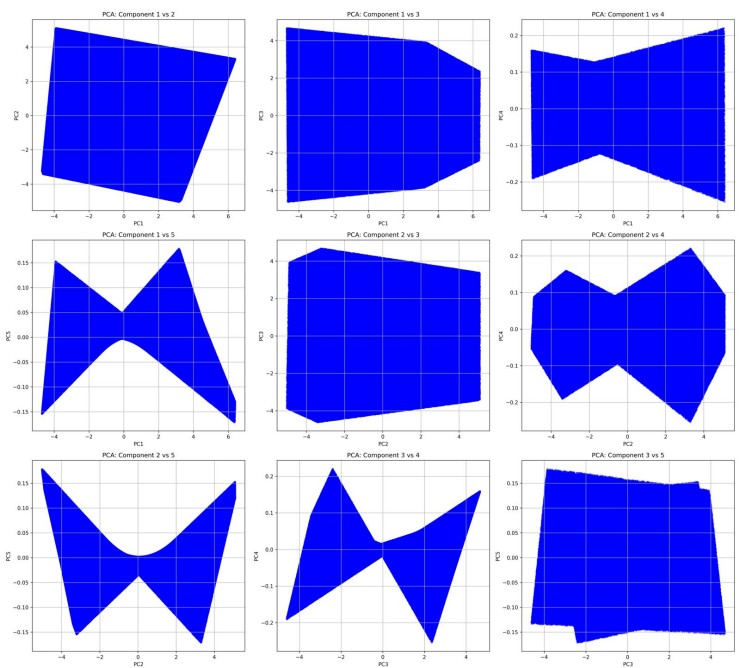

Figure 9: 2D PCA projections of the first 5 components.

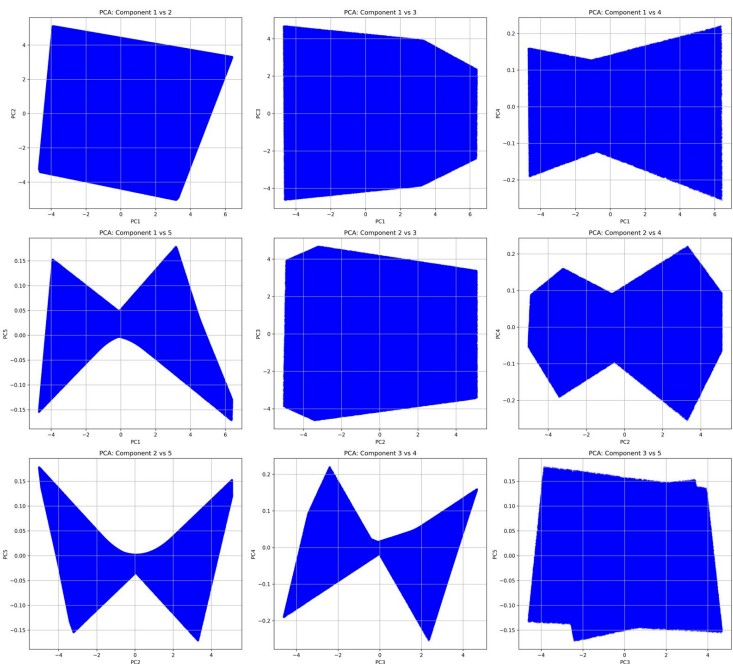

Figure 10: 2D Random Projections of the Data.

# I Experimental Evaluation: Single MDP Comparison

To assess the performance of our proposed binary search algorithm for robust policy evaluation under $L_2$-norm bounded uncertainty, we conduct a series of experiments comparing it against existing methods on fixed Markov Decision Process (MDP) instances. The primary objective is to evaluate convergence speed, accuracy relative to an estimated worst-case value, and consistency across different problem configurations. More details of these experiments along with others can

be found in the appendix, and codes are available at `https://anonymous.4open.science/r/Kernel-Robust-RL-B742/`

## I.1 Experimental Setup

**Algorithms Compared**    We evaluate the following algorithms:

1. **Our Method:** The binary search algorithm presented in this work, which leverages a spectral method for computing the key bisection function $F(\lambda)$.
2. **CPI (Frank-Wolfe):** The Conservative Policy Iteration algorithm adapted from [21] for general robust policy evaluation.
3. **SA-Rectangular $L_2$ VI:** Robust Value Iteration for (s,a)-rectangular $L_2$ uncertainty, a common baseline representing a structured relaxation.
4. **S-Rectangular $L_2$ VI:** Robust Value Iteration for (s)-rectangular $L_2$ uncertainty, another structured relaxation.

**Benchmark Generation**    For each MDP instance and policy, we establish an empirical benchmark for the worst-case robust value. This is achieved by sampling 1,000 transition kernels from the $L_2$ ball of radius $\beta$ centered at the nominal kernel $P_{\text{nominal}}$. Each sampled kernel is projected to ensure it remains a valid stochastic matrix and stays within the $L_2$ ball. The policy $\pi$ is evaluated for each sampled kernel, and the minimum value obtained across these samples, $V_{\text{benchmark}}^{\text{min}}$, serves as our reference robust value.

**MDP and Policy Configuration**    Experiments are conducted on randomly generated MDPs. For each trial, a nominal transition kernel, a reward function, and a uniform initial state distribution $\mu$ are generated. A fixed, randomly generated stochastic policy $\pi$ is then used for robust policy evaluation by all algorithms.

**Experimental Configurations**    Two main sets of single MDP comparisons are performed:

1. Varying State Space ($S$): $S \in \{10, 50, 100, 200\}$, with actions $A = 10$ and uncertainty radius $\beta = 0.01$.
2. Varying Uncertainty Radius ($\beta$): $\beta \in \{0.005, 0.01, 0.05, 0.1\}$, with state space $S = 100$ and actions $A = 10$.

The discount factor is $\gamma = 0.9$. Algorithms are run until convergence (tolerance of $10^{-6}$) or a maximum iteration limit (100).

## I.2 Results and Discussion

Figures 11 and 12 present the convergence behavior of the evaluated algorithms on representative MDP instances for the varying state space and varying uncertainty radius configurations, respectively. Each subplot shows the estimated robust value versus algorithm iterations. The horizontal dashed line indicates $V_{\text{benchmark}}^{\text{min}}$. An algorithm's final point is marked with a star ($\star$) if its estimated robust value converges to within $10^{-6}$ of $V_{\text{benchmark}}^{\text{min}}$.

**Observations**

- **Convergence Speed and Accuracy of Our Method:** Across all tested configurations, Our Method consistently demonstrates superior performance. It generally converges in fewer iterations and achieves a final robust value remarkably close to $V_{\text{benchmark}}^{\text{min}}$, as frequently indicated by the star marker. This suggests efficient and accurate identification of the robust penalty $\lambda^*$.
- **CPI Performance:** The CPI algorithm typically converges but often settles at a value slightly higher (less pessimistic) than $V_{\text{benchmark}}^{\text{min}}$. While providing a robust estimate, its subproblem, in the version tested, explores extreme points of the set of all stochastic kernels, which may not always precisely align with the worst-case kernel strictly within the $L_2$ ball.

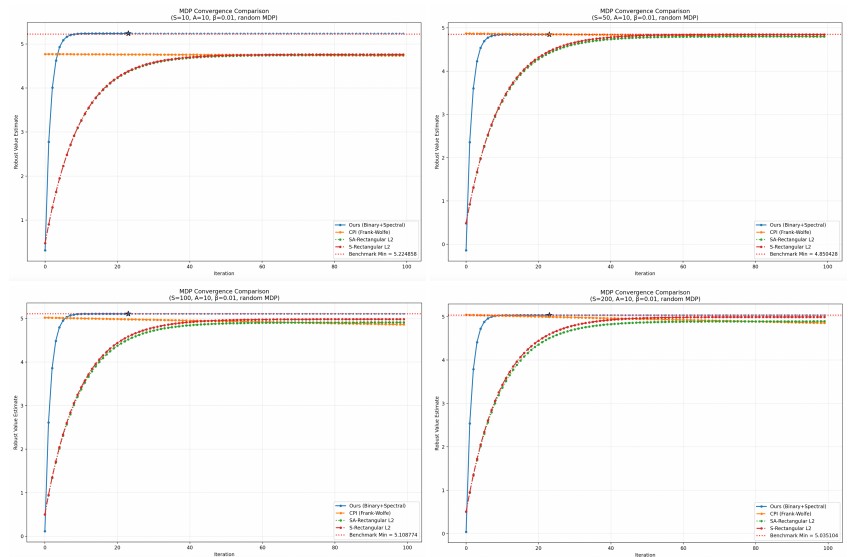

Figure 11: Convergence of robust policy evaluation algorithms for varying state space sizes ($S$). Algorithms whose final value is within $10^{-6}$ of the benchmark are marked with a star ($\star$)

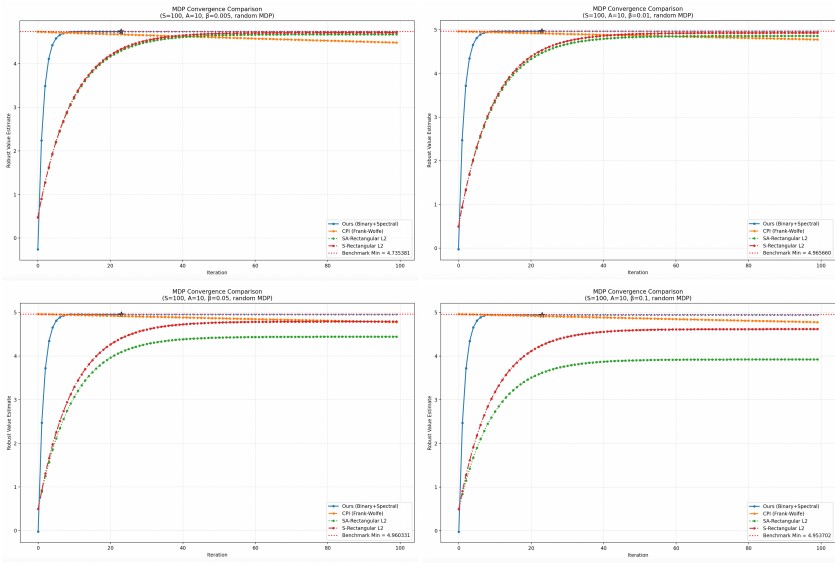

Figure 12: Convergence of robust policy evaluation algorithms for varying uncertainty radius ($\beta$). Algorithms whose final value is within $10^{-6}$ of the benchmark are marked with a star ($\star$)

- **Rectangular Relaxations:** Both sa-rectangular and s-rectangular $L_2$ VI methods consistently converge to robust values significantly lower than those found by Our Method, CPI, and $V_{\text{benchmark}}^{\min}$. This highlights the conservatism inherent in rectangular relaxations when dealing with non-rectangular uncertainty.

- **Consistency Across Setups:** The advantages of Our Method in terms of faster and more accurate convergence are maintained robustly across different state space sizes and uncertainty radius.

## I.3 Performance with large uncertainty radius.

Our theory is limited to only small enough radius, however we show that our algorithm works with bigger radius as well. **Experimental Setup**

- **Environment:** Fixed state space $S = 12$, action space $A = 8$, discount factor $\gamma = 0.9$. The nominal kernel $P_0$ and reward function $R$ were generated randomly.
- **Random Sampling:** For each uncertainty radius $\beta$, we sampled over 500 million kernels from the uncertainty set and computed the empirical minimum return.
- **Algorithm 1 (Ours):** For each $\beta$, our algorithm produced the worst-case rank-one perturbation $bk^\top$. We then projected $P_0 - bk^\top$ onto the simplex to obtain a valid kernel $P'$, and reported the return $J_{P'}^\pi$ as the worst-case estimate.
- **Time Taken:** Our algorithm typically takes 2–4 seconds, while random sampling of 500+ million kernels takes 400 minutes for each $\beta$.
- **Nominal Return:** The nominal return was $-0.1414$.

Table 4: Robust Return with different uncertainty raiduses.

| $\beta$ | $10^{-4}$ | $10^{-3}$ | 0.002 | 0.01 | 0.05 | 0.1 | 0.5 | 1.0 | 2.0 |
|---|---|---|---|---|---|---|---|---|---|
| **Algorithm 1 (Ours)** | -0.141 | -0.142 | -0.143 | -0.150 | -0.186 | -0.231 | -0.588 | -1.026 | -1.851 |
| **Random Sampling** | -0.141 | -0.142 | -0.142 | -0.145 | -0.159 | -0.176 | -0.315 | -0.467 | -0.779 |

**Observation**

As seen from the table, our algorithm consistently identifies worst-case kernels yielding significantly lower (i.e., more conservative) returns compared to the empirical minimum over 500 million random samples—even for large uncertainty radii. Furthermore, our method works much faster than random sampling.

**Conclusion**

Our method performs exactly as expected for small uncertainty sets, with theoretical guarantees. For larger uncertainty radii—where theory no longer applies directly—our algorithm still significantly outperforms random sampling in estimating worst-case returns. We believe this demonstrates the robustness and practical utility of our approach beyond the theoretically guaranteed regime.

*System:* AWS EC2 instance: c7a.32xlarge **vCPUs:** 128 **Memory:** 256 GiB **Network Bandwidth:** Up to 50 Gbps **Processor:** 4th generation AMD EPYC **Clock Speed:** Up to 3.7 GHz.

## J  Experiments: Robust Policy Evaluation with $L_2$ norm

We conduct a numerical comparison of our Algorithm 1 and CPI (Algorithm 3.2 from [21], reproduced as Algorithm 2 in the appendix) for robust policy evaluation. The experiments are performed using a randomly generated nominal kernel $\hat{P}$, reward function $R$, and policy $\pi$. An uncertainty set $\mathcal{U}_2$ is constructed using the nominal kernel with a fixed uncertainty radius $\beta$.

---

**Algorithm 5** Spectral method for computing $\max_{x \in \mathcal{B}} \|Ax\|_2$

---

1: Compute eigenvector $v_i$ and eigenvalues $\lambda_i$ of $A^\top A$

2: WLOG let $\|v_i^+\|_2 \geq \|v_i^-\|_2$ where $v_i^+ = \max(v_i, 0)$, $v_i^- = -\min(v_i, 0)$

3: Compute best score : $j = \arg\max_i \lambda_i \langle v_i, \frac{v_i^+}{\|v_i^+\|_2} \rangle$.

4: **Output:** Approximate maximum value $\beta \|A \frac{v_j^+}{\|v_j^+\|_2}\|_2$.

---

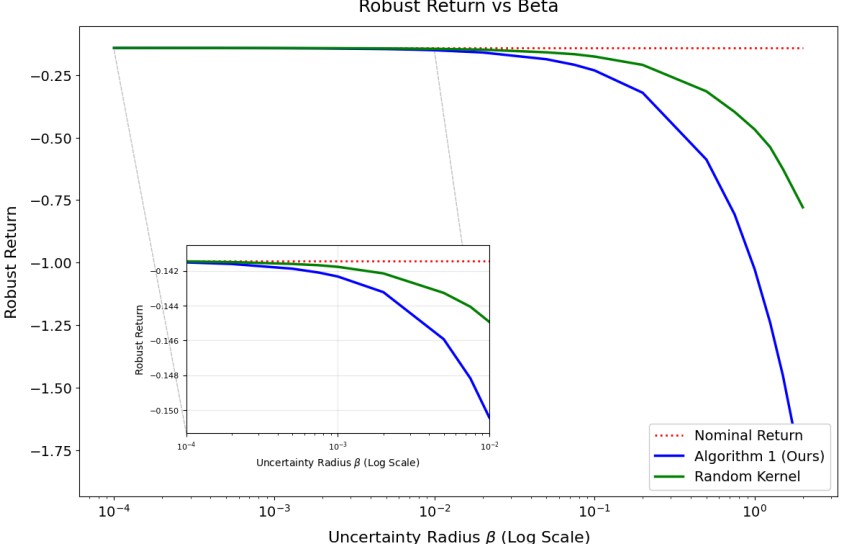

Figure 13: Robust Return for $\beta$ (Uncertainty Radius, S=12 and A=8, $\gamma$=0.9 and convergence tolerance of $10^{-4}$ . Robust Return of Random Kernel is minimum of 500 millions kernels samples for each $\beta$

Figure 14 demonstrates the convergence behavior of both methods, presenting results based on the number of iterations (left panel) and computation time (right panel). The left panel shows the robust return achieved per iteration, while the right panel depicts the robust return as a function of wall-clock time. Note that the x-axes of the figure have a logarithmic scale in order to clearly capture the slow convergence of the CPI method.

- **Our Algorithm 1.** We apply our Binary Search Algorithm 1 to perform robust policy evaluation with the given nominal kernel $\hat{P}$ and uncertainty radius $\beta$. Each iteration of the algorithm involves computing $F(\lambda)$, for which our Spectral Algorithm 5 is employed. Our algorithm converges very quickly requiring only a few iterations.

- **Algorithm 3.2 of [21].** We run Algorithm 2 with precomputed values of $d^\pi$ and $A^\pi$. The step sizes are chosen to be either a small constant or dynamically adjusted, as described in the algorithm. Note that Line 3 of the algorithm involves solving $\arg\min_{P \in \mathcal{U}_2} \langle x, P \rangle$. This constrained optimization is solved using a numerical method (scipy.minimize). This gradient based method improves very slowly and converges very far from the true robust return as the uncertainty set $\mathcal{U}_2$ is very non-rectangular.

- **Brute Force Benchmark.** To approximate the true robust return, we generate a large number of random samples $\{P_i \mid i \leq n\}$ from $\mathcal{U}_2$ and estimate the empirical minimum, $\min_i J^\pi_{P_i}$, as a proxy for the robust return. Note this method requires exponential number of samples to reasonably cover the entire uncertainty set. Hence the values obtained in Figure 14, are an approximate upper bound on the true robust return.

The results in Figure 14 reflect a general trend observed across a wide range of experiments conducted with state space sizes ranging from $S = 5$ to $S = 190$ and uncertainty radius $\beta \in \{0.005, 0.01, 0.05\}$. Our proposed algorithm consistently demonstrates superior performance, converging in significantly fewer iterations and less computation time while the computational demands of the CPI algorithm grow substantially with larger state spaces. Hence, our method exhibits more favorable scaling properties, making it practical for high-dimensional problems.

The codes, detailed explanations, and additional experiments are available at `https://anonymous.4open.science/r/non-rectangular-rmdp-77B8`. System details for the experiments are as follows: **Operating System:** macOS Sequoia (Version 15.4.1), **Chip:** Apple M2, **Cores:** 8 (4 performance and 4 efficiency), **Memory:** 16 GB (LPDDR5).

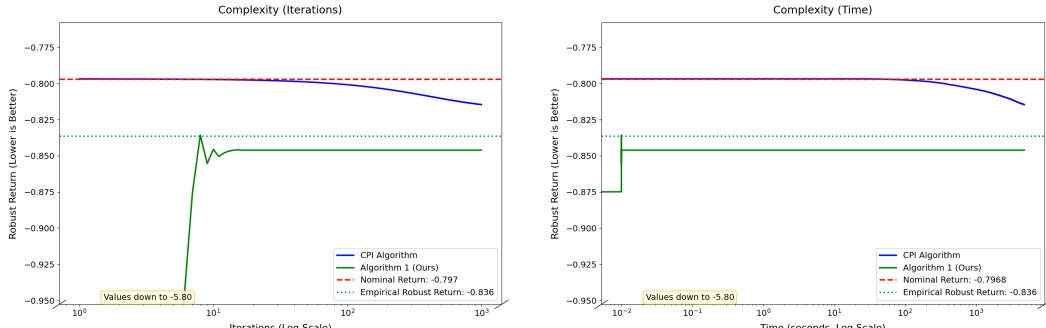

Figure 14: Comparison of Algorithm 1 (Ours) and the CPI Algorithm for $\beta$ (Uncertainty Radius) $= 0.05$, $S = 10$, $A = 8$, $\gamma = 0.9$, and a convergence tolerance of $10^{-4}$.

