# OpenReview forum: "Non-rectangular Robust MDPs with Normed  Uncertainty Sets"
_NeurIPS.cc/2025/Conference — NeurIPS 2025 poster_

### Official Review · Reviewer_vgZb · 2025-06-20

**Clarity:** 2
**Significance:** 3
**Originality:** 3
**Rating:** 4
**Confidence:** 5

**Summary:**

The paper studies policy evaluation problems of non-rectangular robust MDPs with normed uncertainty sets. The key technical ingredient is the use of the Sherman-Morrison formula to reformulate the objectives and derive the fixed point equation $F(\lambda)$. Because of them, the proposed bisection algorithm can achieve $\log(\epsilon^{-1})$ convergence rate.

**Questions:**

Can the method be generalized to other types of uncertainty sets? e.g. $ \{P:\sum_{s,a} D(P_{s,a},\hat{P}_{s,a})\leq \delta\} $, where $D$ could be some divergences or Wasserstein distance.

**Ethical Concerns:**

["NO or VERY MINOR ethics concerns only"]

**Final Justification:**

The paper is solid. But it has a limited scope. Therefore, I maintain my score as borderline accept.

**Limitations:**

The paper only considers policy evaluation. Policy learning is still unclear.

It is known that in the robust MDPs, history-dependent policies may achieve strictly higher rewards than stationary policies. However, how to evaluate a history-dependent policy is not discussed in this paper.

**Quality:**

3

**Strengths And Weaknesses:**

Most of the existing literature on robust MDPs focuses on the rectangular uncertainty sets because of their tractability. However, non-rectangular robust MDPs may be less conservative. Therefore, I believe the contributions of this paper is siginificant, which addresses computational challenges of policy evaluations of robust MDPs with a special class of uncertainty sets.
In particular, the use of the Sherman-Morrison formula and the derivation the fixed point equation $F(\lambda)$ is smart and useful.

However, I believe the presentation of the paper can be further improved:

* The notation is heavy and not always used carefully. For example, $D^\pi$ is introduced on line 82 without a subscript, and $D^\pi_P$ is never formally defined.
* On line 121, the expression should be written using double bars, i.e., $\left( \sum_{s,a,s'} |P(s'|s,a) - \hat{P}(s'|s,a)|^p \right)^{1/p}$.
* The policy-averaging linear operator $H^\pi$ first appears on line 212 but is not defined until line 227.

The authors may also consider citing Section 6.9 in [1] for the dual formulation introduced on line 101.

[1] Puterman, Martin L. Markov decision processes: discrete stochastic dynamic programming. John Wiley & Sons, 2014.

---

> ### Author Rebuttal · Authors · 2025-07-31
>
> We thank the reviewer for recognizing the significance of moving beyond rectangular uncertainty sets. We're glad our use of the Sherman–Morrison formula and fixed-point formulation was found useful, as these were central to achieving both tractability and insight in our analysis.
>
> We welcome the reviever feedback on the presentation that we fixed in the draft and it will reflect in the final version. We answer the major questions below.
>
> ***Q1) Can the method be generalized to other types of uncertainty sets?***
> > The current method is specifically designed for $L_p$ robust MDPs, relying critically on the rank-one perturbation structure of the worst-case kernel, which enables closed-form matrix inversion.
>
> > However, we believe that this approach could also be extended to KL-divergence-based uncertainty sets. As shown in [30], the worst-case kernel under KL divergence has an exponentiated closed form. If a closed-form or practical approximation for the associated matrix inverse can be derived, it would lead to both useful and insightful extensions of the method.
>
>
>
> ***Q2) The paper only considers policy evaluation. Policy learning is still unclear.***
>
> > Policy gradient methods for non-rectangular robust MDPs have been shown to converge to a globally optimal solution, with an iteration complexity of $O(\epsilon^{-4})$ [32]. However, this result assumes oracle access to the exact policy gradient, which in turn requires accurate policy evaluation.
>
> > In summary, once robust policy evaluation is performed, policy improvement via policy gradient can be carried out efficiently.
>
>
> ***Q3) It is known that in the robust MDPs, history-dependent policies may achieve strictly higher rewards than stationary policies. However, how to evaluate a history-dependent policy is not discussed in this paper.***
> >  Yes, we apologize for this omission. However, robust policy evaluation has been proven to be strongly NP-hard even for stationary policies [35], making the extension to non-stationary policies considerably more complex and unwieldy. In other words, robust evaluation for non-stationary policies lies beyond the scope of this work. We will include this point as a limitation in the final version.

---

> > ### Comment · Reviewer_vgZb · 2025-08-01
> > **Thanks for the response**
> >
> > Thank authors for the response. For Q2, I am still unsure whether the gradient of the policy can be easily obtained from the robust policy evaluation procedures described in this paper.

---

> > > ### Author Response · Authors · 2025-08-01
> > > **Evaluation of robust gradient**
> > >
> > > We thank the reviewer for the prompt response.
> > >
> > > ***Regarding Q2)***  Our algorithm can output a estimated worst kernel along with robus return.  Precisely, our algorithm can out put rank one perturbation $bk^\top$ and then we can project $P_0 -bk^\top$ onto simplex to get the estimated worst kernel $P'$.
> > >
> > > We can compute the policy gradient ($d^\pi_{P'}•Q^\pi_{P'}) $ w.r.t to this kernel, in the robust policy gradient in [32]. Note that [32] requires only estimated worst kernel for the convergence .
> > >
> > > We would be glad answer to other questions,  if any.

---

> > > > ### Comment · Reviewer_vgZb · 2025-08-04
> > > > **Thanks for your reply**
> > > >
> > > > Thanks for your reply. Now I understand the paper’s contribution better.

---

### Official Review · Reviewer_p4ao · 2025-06-26

**Clarity:** 3
**Significance:** 2
**Originality:** 2
**Rating:** 3
**Confidence:** 3

**Summary:**

This technical paper studies the problem of robust policy evaluation (RPE) in robust Markov decision processes (RMDPs) with *non-rectangular* uncertainty sets, for which existing results suffer from either exponential iteration complexity or coarse approximation bounds. This paper identifies a class of $L_p$-bounded uncertainty sets that can break such difficulties. Dual representation and a resulting efficient RPE algorithm with iteration complexity $\log(1/\epsilon)$ are developed.

**Questions:**

1. How small should the uncertainty set radius $\beta$ be to make sure that the non-rectangular set contains valid probability kernels? The paper only emphasizes that such a key parameter should be small, but how small should it be? It would be better to incorporate some quantitative characterization.
2. Also, how does the convergence speed of Algorithm 1 depend on the uncertainty set size $\beta$? This is of interest because one would like to see how the proposed algorithm behaves as the size of the uncertainty set changes.

**Ethical Concerns:**

["NO or VERY MINOR ethics concerns only"]

**Final Justification:**

The reviewer is still concerned with the motivations of the proposed uncertainty set, the novelty of the analysis, as well as the characterization of the convergence rate of the algorithm. The overall goal of the paper has nice merits, but the reviewer believes it is better to go through further revision before acceptance. Therefore, I am maintaining my original score.

**Limitations:**

Please see the **Questions** part.

**Quality:**

3

**Strengths And Weaknesses:**

**Strengths:**
1. The newly proposed uncertainty set is meaningful in terms of a providing a computationally tractable formulation of non-rectangular uncertainty sets for RMDPs.
2. Most of the theoretical results are sound and well developed. The literatures are sufficiently reviewed and the problem setups are clearly introduced.

**Weakness:**
1. I agree that in general develop theory and algorithms of non-rectangular uncertainty sets are important topics, but the explanation provided in this paper is not very convincing to me. The point argued here, as well as the idea of the $L_p$ bounded uncertainty sets, is that across different state action pairs only certain of the transition probabilities could deviate much from the nominal model. This is more like a domain specific prior and might not always hold. Probably a better way to argue this is to provide more concrete mathematical examples of certain application domains to show that this is the case.
2. The theory and the algorithms proposed in this work are relatively direct provided with existing results on rectangular uncertainty sets.

---

> ### Author Rebuttal · Authors · 2025-07-31
>
> We thank the reviewer for their positive feedback.
>
> We’re glad the reviewer finds the proposed uncertainty set meaningful for tractable robust MDP formulations. We also appreciate the recognition of the soundness of our theoretical results and the clarity of our literature review and problem setup.
>
> ***Q1) I agree that in general develop theory and algorithms of non-rectangular uncertainty sets are important topics, but the explanation provided in this paper is not very convincing to me. The point argued here, as well as the idea of the $L_p$ bounded uncertainty sets, is that across different state action pairs only certain of the transition probabilities could deviate much from the nominal model. This is more like a domain specific prior and might not always hold. Probably a better way to argue this is to provide more concrete mathematical examples of certain application domains to show that this is the case.***
> > There are several natural ways where $L_p$ bounded uncertainty sets arise. For example, if the model is the combination of a nominal model + noise, and if the noise has $L_P$ shape this naturally leads to an $L_p$ shape, cf. [21].
>
> ***Q2) The theory and the algorithms proposed in this work are relatively direct provided with existing results on rectangular uncertainty sets.***
> >Yes, the reviewer is correct in observing that our work builds upon prior research on rectangular robust MDPs. Nonetheless, we believe that the contributions and insights presented in this work remain significant and stand on their own merit.
>
> ***Q3) How small should the uncertainty set radius $\beta$ be to make sure that the non-rectangular set contains valid probability kernels? The paper only emphasizes that such a key parameter should be small, but how small should it be? It would be better to incorporate some quantitative characterization.***
>
> > As the uncertainty radius increases, the worst-case transition kernel $P^*$ may no longer admit a rank-one perturbation form of the nominal kernel. In such cases, the candidate output from our algorithm, $P_0 - bk^\top$, may fall outside the probability simplex and thus may not represent a valid transition kernel. To address this, we project $P_0 - bk^\top$ onto the simplex to obtain a valid transition kernel $P'$.
>
> > Furthermore, while the projected kernel $P'$ may not maintain the rank-one structure, we expect it to be close to the true worst-case kernel $P^*$. To support this, we conducted a new experiment comparing our method with random sampling across various uncertainty radii.
>
> ### Experimental Setup
>
> - **Environment**: Fixed state space $S = 12$, action space $A = 8$, discount factor $\gamma = 0.9$. The nominal kernel $P_0$ and reward function $R$ were generated randomly.
> - **Random Sampling**: For each uncertainty radius $\beta$, we sampled over 500 million kernels from the uncertainty set and computed the empirical minimum return.
> - **Algorithm 1 (Ours)**: For each $\beta$, our algorithm produced the worst-case rank-one perturbation $bk^\top$. We then projected $P_0 - bk^\top$ onto the simplex to obtain a valid kernel $P'$, and reported the return $J^\pi_{P'}$ as the worst-case estimate.
> - **Time Taken**: Our algorithm typically takes $2-4$ seconds  while random sampling of 500+ million random sampling takes 400 minutes for each $\beta$.
> - **Nominal Return**: The nominal return was $-0.1414$.
>
> ### Results
>
> | **$\beta$**             | **0.0001** | **0.0005** | **0.001** | **0.002** | **0.01** | **0.05** | **0.1**  | **0.5**  | **1.0**   | **2.0**   |
> |-------------------------|:----------:|:----------:|:---------:|:---------:|:--------:|:--------:|:--------:|:--------:|:---------:|:---------:|
> | **Algorithm 1 (Ours)**  | -0.1415    | -0.1419    | -0.1423   | -0.1432   | -0.1504  | -0.1864  | -0.2312  | -0.5878  | -1.0259   | -1.8509   |
> | **Random Sampling**     | -0.1415    | -0.1416    | -0.1418   | -0.1421   | -0.1449  | -0.1589  | -0.1757  | -0.3149  | -0.4671   | -0.7788   |
>
> ### Observation
>
> As seen from the table, our algorithm consistently identifies worst-case kernels yielding significantly lower (i.e., more conservative) returns compared to the empirical minimum over 500 million random samples—even for large uncertainty radii. And our method is much faster than random sampling by order of magnitudes.
>
> ### Conclusion
>
> Our method performs exactly as expected for small uncertainty sets, with theoretical guarantees. For larger uncertainty radii—where theory no longer applies directly—our algorithm still significantly outperforms random sampling in estimating worst-case returns. We believe this demonstrates the robustness and practical utility of our approach beyond the theoretically guaranteed regime.
>
>
> ***System***: aws ec2 instance: c7a.32xlarge
> vCPUs: 128
> Memory: 256 GiB
> Network Bandwidth: Up to 50 Gbps
> Processor: 4th generation AMD EPYC
> Clock Speed: Up to 3.7 GHz
>
> ***Q4) Also, how does the convergence speed of Algorithm 1 depend on the uncertainty set size $\beta$? This is of interest because one would like to see how the proposed algorithm behaves as the size of the uncertainty set changes.***
> > The theoretical convergence rate of Algorithm 1 is independent of the uncertainty radius $\beta$. Its empirical performance in terms of accuracy, as $\beta$ increases, is presented in the table above.

---

> > ### Comment · Reviewer_p4ao · 2025-08-03
> >
> > Thank you for your response and the additional experiments. However, the reviewer is still concerned with the motivations of the proposed uncertainty set, the novelty of the analysis, as well as the characterization of the convergence rate of the algorithm. The overall goal of the paper has nice merits, but I believe it is better to go through further revision before acceptance. Therefore, I choose to maintain my original score.

---

> > > ### Author Response · Authors · 2025-08-06
> > >
> > > We are glad the reviewer appreciates the overall mertis of the paper. We answer below concerns raised by the reviewer.
> > >
> > >
> > >  ***C1) Motiavtion of the proposed uncertainty sets.***
> > >
> > > >***Rectangular uncertainty set are tractable but not practical***. Almost all the uncertainty sets considered in the literartue are assumed to be rectangular. And this rectangular assumption is not motivated by its practical significance rather its computational tractibility. In other words, there are almost no reason  to  believe that the uncertainty sets arising in practice would be rectangular. There are very few works on the non-rectangular uncertainty sets because of existing NP-Hardness result.
> > >
> > > > ***Rectangular uncertainty sets require atleast $S$ uncertainty radius parameters**** which can be in-efficient to compute or guess in the first place. On the other hand, our non-rectanguar $L_p$ bounded uncertainty set has only one pararmer which is much easier to work with.
> > >
> > > >***Case studies***: Suppose we have large enough offline samples to make empirical model of the environment. Unless the data is independently sampled in each state with reset, the the empirical model is going to be non-rectangular, as most of the time a fixed policy (or few of them) is employed to generate the data. Another example could be sim-to-real, where simulator can't be seen to deviate from the reality in rectangular form.
> > >
> > > > ***Other bounding norm***: Yes, it is possible that KL or some other distance captures the uncertainty better in some scenarios than $L_p$. Nonetheless, this is the first tractable method for any non-rectangular uncertainty sets which can be seen as a good starting point for other cases.
> > >
> > > ***C2)Novelty of the analysis.***
> > >
> > > ***Simple maths yielding good results is a strength.*** We agree that our paper builds heavily on the previous rectangular works. And we use simple mathematics yet it generates very non-trivial findings, hence we believe  it should be seen as strength rather than weakness.
> > >
> > > ***C3) Convergence rate.***
> > >
> > >  Algorithm 1 convergence in $O(\log(\frac{1}{\epsilon}))$ as its a binary-search method for all $p$. We refer to the Proposition F.1 and Lemma F.2 for its elegant proof.
> > >
> > >  Further, Algorithm 1 requires solving a matrix norm as subroutine whose complexity is given below:
> > > >***For $p=1$***, as shown in the response to reviewer oaPp, the sub-problem can be solved in the close form that requires the access to standard quanties such as occupation measure, value function w.r.t. nominal kernel etc. Hence, its complexity is $O(S^2A\log(\frac{1}{\epsilon}))$.
> > >
> > > > ***For $p=\infty$***,  our non-rectangular is equivalent to sa-rectangular case, which is already efficiently solvable [19].
> > >
> > > > ***For $p=2$***, we provided heuristic spectral Algorithm 2 whose comlexity is $O(S^3A^3)$.
> > >
> > > > ***For $p\neq 1,2,\infty$***, we are not aware of any positive or negative results for our sub-problem evaluation.
> > >
> > > To summarize the complexity for non-rectangular robust policy evaluation is $O(S^2A\log(\frac{1}{\epsilon}))$ for $L_1$ bounded uncertainty set to achieve $\epsilon$-close worst kernel. And for $p=2$, the efficiency of our method can be empirically be seen the table provided in previous response.
> > >
> > > We will gladly emphasize upon this  in the final version.

---

> > > > ### Author Response · Authors · 2025-08-07
> > > >
> > > > Given the approaching deadline, we would be very happy to address any last minute questions, if any.

---

### Official Review · Reviewer_5k8B · 2025-07-01

**Clarity:** 2
**Significance:** 3
**Originality:** 3
**Rating:** 4
**Confidence:** 3

**Summary:**

This paper discusses a subclass of non-rectangular Robust Markov Decision Processes (RMDPs) for which the NP-hardness result of the policy evaluation problem [1] does not hold. This subclass is RMDPs with L_p ball uncertainty sets around a nominal kernel.
They first argue that assuming s- or s,a-rectangularity can lead to high conservativeness in cases where the uncertainty set is actually non-rectangular. Next, they show that for the RMDP subclass with L_p ball uncertainty sets with p>1, the NP-hardness result does not hold, since this result relies on a polyhedral structure in the uncertainty set of the RMDP.
After showing that the NP-hardness result does not hold, they show that L_p balls with p>1 are equivalent to the union of an infinite number of s,a-rectangular sets. They use this observation to construct a new dual formulation for the policy evaluation problem, which they show can be solved efficiently with a binary-search based algorithm.
They compare their algorithm for the L_2 ball to the Conservative Policy Iteration (CPI) algorithm of [2]. These experiments show that their dedicated algorithm outperforms the CPI algorithm.


[1] Wolfram Wiesemann, Daniel Kuhn, and Berç Rustem. Robust Markov decision processes. Math. Oper. Res., 38(1):153–183 (2013).

[2] Mengmeng Li, Daniel Kuhn, and Tobias Sutter. Policy Gradient Algorithms for Robust MDPs with Non-Rectangular Uncertainty Sets. CoRR abs/2305.19004 (2023).

**Questions:**

- Q1) What is the justification for considering memoryless randomized policies, while [1] states that for non-rectangular uncertainty sets both memory and randomization are needed?
- Q2) Where do the negative exponents come from in proposition 3.1 and on line 23? How does this correspond to the positive exponent in line 149?
- Q3) What happens to the algorithm when considering p != 2?

**Ethical Concerns:**

["NO or VERY MINOR ethics concerns only"]

**Final Justification:**

I believe the paper has some merit, and I was happy to read the authors intend to discuss the limitation to policy evaluation of stationary policies in the revised version of the paper, and I believe that the evaluation of stationary policies is a valuable and interesting starting point.
Although I still think that the evaluation of the algorithm and the discussion of p!=2 are limited, I believe the theoretical contribution is sufficiently interesting to raise my score to a borderline accept, in particular with the perspective of future research in the non-rectangular RMDP direction.

**Limitations:**

Yes.

**Paper Formatting Concerns:**

None.

**Quality:**

2

**Strengths And Weaknesses:**

## Strengths

- The main strengths of this paper are the insight that the NP-hardness result of [1] does not necessarily apply to L_p ball uncertainty sets and the notion that there might be more uncertainty sets with this property. This work therefore exposes an interesting direction for future research.

- Additionally, although I do not have the background to understand all the details, the discussion of the dual formulation seems technically solid.

- Finally, the binary search algorithm proposed in this paper strongly outperforms the CPI algorithm in the experiment that is discussed in the main paper.


## Weaknesses

My main concerns with the paper are two simplifications that are not clearly justified in the main paper. First of all, the work only considers memoryless randomized agent policies, while the cited work of Wieseman [1] states that optimal agent policies in non-rectangular RMDPs require both randomization and memory. Although one of the main points of this paper is that the NP-hardness result from [1] does not translate to their setting, I do not see any arguments as to why this simpler policy type is appropriate now.
Secondly, Algorithm 2 is only discussed for L_2 ball uncertainty sets (so p=2), and no explanation is given on how a different value for p would change the algorithm or influence the complexity results. The experiments are also done only for L_2 balls.

Furthermore, the experiments and discussion thereof are limited. The authors state that they have conducted multiple experiments, but only present the results of a single experiment in the main paper, claiming that the other experiments show the same trend. The results of the rest of the experiments are not in the appendix, and the anonymous GitHub repository that contains the "additional experiments" according to the paper only contains the code to run the additional experiments yourself.
Additionally, the paper does not compare their algorithm against any rectangular methods, while part of this paper is dedicated to discussing the overconservativeness of assuming rectangularity for a non-rectangular uncertainty set. Such a comparison would be valuable and expected.

Moreover, Proposition 3.1 is unclear to me. The negative exponents in the complexity of the volume ratios would imply that the complexity becomes smaller as S and A become larger. This also doesn't correspond to the complexity given in line 149, which has a positive exponent, although a version with the negative exponent is written in line 23.
Another note on Proposition 3.1 is that it is based on s- and s,a- rectangular sets that enclose the L_2 ball. However, the L_2 ball is also an approximation of the uncertainty set according to the pictures in Figure 1. It might therefore be possible that the s- or s,a- rectangular sets that encase the uncertainty do not need to encase the entire L_2 ball. This is not discussed in the paper.

Finally, the related work is discussed in limited detail. In particular, the authors summarize the existing research on RMDPs in just a single sentence with 18 citations that are not further explained, and then immediately move on to the non-rectangular RMDPs. There is a bit more discussion in the appendix, but this should be part of the main paper.

## Minor remarks
- line 20: "non-rectangular uncertainty set could be thought of as ..." plural vs singular.
- line 35: "given oracle access to the robust gradient ..." word ("a") missing, I think.
- line 75: "..., and U set of transition kernel P ..." word ("a") missing and singular should be plural ("kernels"), I think.
- line 96: "A L_p-bounded uncertainty sets ..." plural vs singular.
- Figure 1: The left image is labeled 3D, but this should be 2D.
- The Notes and Definitions section after the NeurIPS checklist does not seem a proper part of the appendix. There is no letter in front.
- Figure 2: Caption missing.
- References to the proofs/content in the appendix are missing. (line 177, lines 291-296)
- The answer to the NeurIPS checklist question 9 is missing, but from the justification, it is clear that this should be a [NA]
- The bibliography is inconsistent. It is not possible to quickly see whether the cited papers have been published somewhere and where.

---

> ### Author Rebuttal · Authors · 2025-07-31
>
> We thank the reviewer for their thoughtful and encouraging feedback.
>
> We’re glad the reviewer found the insight—that NP-hardness need not apply to $L_p$ ball uncertainty—interesting, and we appreciate the recognition of its potential to open new research directions. We also thank the reviewer for their positive comments on the dual formulation and the empirical strength of our binary search algorithm.
>
> ***Q1) What is the justification for considering memoryless randomized policies, while [1] states that for non-rectangular uncertainty sets both memory and randomization are needed?***
> > Yes, the reviewer is correct in noting that the optimal robust policy for non-rectangular robust MDPs can, in general, be non-stationary.
>
> > However, it is important to highlight that even robust policy evaluation for *stationary* policies in non-rectangular robust MDPs is known to be **strongly NP-hard**.
>
> > Given the current state of the art, handling non-stationary policies appears significantly more complex and unwieldy, whereas focusing on stationary policies already presents substantial challenges.
>
> > Therefore, we believe that restricting attention to stationary policies is a natural and meaningful starting point. Extending the analysis to non-stationary robust policies remains an important and interesting direction for future work.
>
> ***Q2) Where do the negative exponents come from in proposition 3.1 and on line 23? How does this correspond to the positive exponent in line 149?***
> > We thank the reviewer for pointing this out. The correction has been made in line 149, where the exponent is now appropriately written as negative.
>
> ***Q3) What happens to the algorithm when considering p != 2?***
> > Our result and algorithm hold for all $p \geq 1$. However, for the specific case of $p = 2$, we can efficiently evaluate $F(\lambda)$ using a fast spectral method (Algorithm 2). For other values of $p$, one may need to rely on numerical methods, which could be comparatively slower.
>
> ***Q4) Other minor comments.***
> > We sincerely thank the reviewer for the invaluable time and effort dedicated to reviewing our paper. We appreciate the constructive comments, most of which we have already addressed and will be reflected in the final version.

---

> > ### Comment · Reviewer_5k8B · 2025-08-04
> >
> > Thank you for the clarifying answers to my questions.
> >
> > I was happy to read, although in the response to another review, that you intend to discuss the limitation to policy evaluation of stationary policies in the revised version of the paper, and I agree that the evaluation of stationary policies is a valuable and interesting starting point.
> >
> > Although I still think that the evaluation of the algorithm and the discussion in case $p\neq2$ are limited, I believe the theoretical contribution is sufficiently interesting to raise my score to a borderline accept, in particular with the perspective of future research in the non-rectangular RMDP direction.

---

### Official Review · Reviewer_oaPp · 2025-07-02

**Clarity:** 3
**Significance:** 3
**Originality:** 3
**Rating:** 5
**Confidence:** 3

**Summary:**

The paper shows that robust policy evaluation for non‑rectangular transition uncertainty defined by an $L_p$ ball is polynomial‑time tractable, in contrast to the known NP‑hardness for general nonrectangular sets . By (i) decomposing the ball into an infinite union of sa‑rectangular sets, (ii) proving the worst‑case kernel is a rank‑one perturbation, and (iii) deriving a dual fixed‑point that can be solved by binary search, the authors obtain an $O(log 1/\epsilon)$ algorithm. Experiments on small-scale random MDPs confirm large speed‑ups over the CPI baseline of prior art [20] Li et al. 2024.

**Questions:**

- Your method requires the radius to be small enough that all perturbed rows remain valid probability vectors. In realistic safety‑critical settings, large shocks or zero‑probability transitions can occur. Could you quantify how performance or guarantees degrade as the radius grows, and comment on whether this limitation is less severe than the approximation gap in [20]? You described Li et al.​ 2024 as “meaningless” because its approximation bound is loose in certain worst‑case instances; then the methods described in the paper should be held to the same standard.

**Ethical Concerns:**

["NO or VERY MINOR ethics concerns only"]

**Final Justification:**

The paper studies an important problem with non-trivial arguments that exploits the impossibility result to its core.

**Limitations:**

yes

**Paper Formatting Concerns:**

None of any major formatting issues are noticed.

**Quality:**

3

**Strengths And Weaknesses:**

Strengths

- The paper is the first to give a provably polynomial-time evaluator for a coupled transition-uncertainty set, namely an Lp ball. This closes a long-standing gap between expressive non-rectangular models and the rectangular relaxations used since Iyengar 2005, and it sidesteps the NP-hardness barrier of Wiesemann et al. 2013 by exploiting norm structure rather than polyhedral structure.

- A dual reformulation reduces the min–max over kernels to a one-dimensional fixed point. Coupled with a Sherman–Morrison update, this yields transparent rank-one worst-case kernels, linear convergence of the bisection loop, and an $O(log 1/\epsilon)$ iteration bound—all easy to verify and intuitively linked to the geometry of the Lp ball.

- The evaluator drops the irreducibility and large-diameter assumptions that hampered earlier CPI-style methods, making it applicable to sparse or disconnected MDPs. The authors release full proofs and runnable code; timing studies up to 190 states show speed-ups of orders of magnitude that agree with the theoretical complexity.

Weaknesses

- Validation is limited to random kernels with preset policies; there is no demonstration on a benchmark control task (grid-world, cart-pole, MuJoCo, etc.) where a robust policy derived from the evaluator outperforms rectangular or nominal baselines. Consequently the practical benefit, beyond runtime, remains speculative.

- The guarantees rely on the uncertainty radius being small enough that every perturbed row stays in the simplex. They do not cover KL-divergence, Wasserstein balls, or large uncertainty radius where new transitions appear—settings that are common in distributionally-robust RL and may break the rank-one structure the theory exploits.

---

> ### Author Rebuttal · Authors · 2025-07-31
>
> We thank the reviewer for the detailed and encouraging feedback.
>
> We’re glad the reviewer appreciates our robust policy evaluation method for coupled transition-uncertainty sets, and the significance of closing the gap between expressive non-rectangular models and tractable rectangular analysis. We’re especially encouraged that the clarity of our dual reformulation, the rank-one kernel structure, and the practical improvements over CPI methods came through clearly. We answer below the questions.
>
> ***Q1) Validation is limited to random kernels with preset policies; there is no demonstration on a benchmark control task (grid-world, cart-pole, MuJoCo, etc.) where a robust policy derived from the evaluator outperforms rectangular or nominal baselines. Consequently the practical benefit, beyond runtime, remains speculative.***
> > We agree that evaluating the empirical performance of a robust algorithm—motivated by our theoretical analysis of non-rectangular uncertainty—on moderately complex domains or real-world problems would be valuable for developing practical methods. However, this extension would require substantial additional work to be practically useful and is beyond the scope of the current paper.
>
> > The primary goal of this work is to initiate study into non-rectangular robust MDPs, a class often considered intractable. We hope that our results will serve as a foundation for future research in this direction, ultimately contributing to real-world applications.
>
>
>
> ***Q2) The guarantees rely on the uncertainty radius being small enough that every perturbed row stays in the simplex. They do not cover KL-divergence, Wasserstein balls, or large uncertainty radius where new transitions appear—settings that are common in distributionally-robust RL and may break the rank-one structure the theory exploits.***
> > Yes, the reviewer is correct in noting that our analysis critically relies on the rank-one structure of the perturbation, which holds only when the uncertainty radius is sufficiently small. This rank-one structure may indeed not be preserved in more general settings, such as those described by the reviewer.
>
> > However, in the case of KL-divergence–based uncertainty sets, the structure of the worst-case transition kernel is known [30]—it takes an exponentiated form rather than a rank-one form. It may be possible to derive an analogue of the Sherman-Morrison formula, or a practically useful approximation, for this case.
>
> > The broader point is that our work serves as a step in the right direction, highlighting the possibility that practical algorithms for non-rectangular robust MDPs may indeed exist. We hope this encourages further exploration in this promising area.
>
> ***Q3) Your method requires the radius to be small enough that all perturbed rows remain valid probability vectors. In realistic safety‑critical settings, large shocks or zero‑probability transitions can occur. Could you quantify how performance or guarantees degrade as the radius grows, and comment on whether this limitation is less severe than the approximation gap in [20]? You described Li et al. 2024 as “meaningless” because its approximation bound is loose in certain worst‑case instances; then the methods described in the paper should be held to the same standard.***
> > First, we sincerely apologize if our wording gave the impression that we were describing the work of Li et al. (2024) as “meaningless.” That was not our intention, and we will revise the phrasing accordingly. What we meant to convey is that the worst-case accuracy bound in their work can, in some settings, exceed the maximum possible sub-optimality—rendering the bound vacuous in those cases.
>
> > Regarding the robustness radius: as the uncertainty radius increases, the worst-case transition kernel $P^*$ may no longer admit a rank-one perturbation form of the nominal kernel. In such cases, the candidate output from our algorithm, $P_0 - bk^\top$, may fall outside the probability simplex and thus may not represent a valid transition kernel. To address this, we project $P_0 - bk^\top$ onto the simplex to obtain a valid transition kernel $P'$.
>
> > The robust return then becomes $J^\pi_{P'}$. Crucially, even in the worst-case projection scenario, the sub-optimality gap $J^\pi_{P'} - J^\pi_{P^*}$ is upper bounded by $\frac{2}{1 - \gamma}$, meaning the result remains meaningful. We will include this clarification in the final version.
>
> > Furthermore, while the projected kernel $P'$ may not maintain the rank-one structure, we expect it to be close to the true worst-case kernel $P^*$. To support this, we conducted a new experiment comparing our method with random sampling across various uncertainty radii.
>
> ### Experimental Setup
>
> - **Environment**: Fixed state space $S = 12$, action space $A = 8$, discount factor $\gamma = 0.9$. The nominal kernel $P_0$ and reward function $R$ were generated randomly.
> - **Random Sampling**: For each uncertainty radius $\beta$, we sampled over 500 million kernels from the uncertainty set and computed the empirical minimum return.
> - **Algorithm 1 (Ours)**: For each $\beta$, our algorithm produced the worst-case rank-one perturbation $bk^\top$. We then projected $P_0 - bk^\top$ onto the simplex to obtain a valid kernel $P'$, and reported the return $J^\pi_{P'}$ as the worst-case estimate.
> - **Time Taken**: Our algorithm typically takes $2-4$ seconds  while random sampling of 500+ million random sampling takes 400 minutes for each $\beta$.
> - **Nominal Return**: The nominal return was $-0.1414$.
>
> ### Results
>
> | **$\beta$**             | **0.0001** | **0.0005** | **0.001** | **0.002** | **0.01** | **0.05** | **0.1**  | **0.5**  | **1.0**   | **2.0**   |
> |-------------------------|:----------:|:----------:|:---------:|:---------:|:--------:|:--------:|:--------:|:--------:|:---------:|:---------:|
> | **Algorithm 1 (Ours)**  | -0.1415    | -0.1419    | -0.1423   | -0.1432   | -0.1504  | -0.1864  | -0.2312  | -0.5878  | -1.0259   | -1.8509   |
> | **Random Sampling**     | -0.1415    | -0.1416    | -0.1418   | -0.1421   | -0.1449  | -0.1589  | -0.1757  | -0.3149  | -0.4671   | -0.7788   |
>
> ### Observation
>
> As seen from the table, our algorithm consistently identifies worst-case kernels yielding significantly lower (i.e., more conservative) returns compared to the empirical minimum over 500 million random samples—even for large uncertainty radii. Furthermore, our methods works much faster than random sampling.
>
> ### Conclusion
>
> Our method performs exactly as expected for small uncertainty sets, with theoretical guarantees. For larger uncertainty radii—where theory no longer applies directly—our algorithm still significantly outperforms random sampling in estimating worst-case returns. We believe this demonstrates the robustness and practical utility of our approach beyond the theoretically guaranteed regime.
>
> ***System***: aws ec2 instance: c7a.32xlarge
> vCPUs: 128
> Memory: 256 GiB
> Network Bandwidth: Up to 50 Gbps
> Processor: 4th generation AMD EPYC
> Clock Speed: Up to 3.7 GHz

---

> > ### Comment · Reviewer_oaPp · 2025-08-03
> >
> > Thank you for your response and for the added experiments. I went through other reviews and responses as well. I have one final question. Although I find the paper's idea and the duality arguments nice, the substep of norm maximization still remains NP-hard, and the paper does not seem to provide a provable algorithm for solving it. Could you further elaborate on this if I am missing something here?

---

> ### Author Response · Authors · 2025-08-06
> **Sub-problem is provably efficiently solvable for $p=1$,   trivial for $p=\infty$  and we provided  heuristic for $p=2$ that works well in practice.**
>
> ***Although I find the paper's idea and the duality arguments nice, the substep of norm maximization still remains NP-hard, and the paper does not seem to provide a provable algorithm for solving it. Could you further elaborate on this if I am missing something here?***
> We are glad that the reviewer finds the paper's idea and duality argument nice. And the the reviewer is correct in noticing that we need to solve $$z := \max_{\lVert b\rVert_p\leq1, \lVert k\rVert_p\leq1, b\succeq 0, <k,1>=0}b^\top A k$$
> as sub-routine in Algorithm 1. And as mentioned in Line 984 that it is a bi-linear optimization. And bilinear optimization is NP-Hard in general. But we have a special form, which might be solvable for many cases, as illustrate below.
>
> > ***For  $p=1$:*** For uncertainty case $ \mathcal{U}1$  bounded by $L_1$-ball. We have the close-form solution to our sub-problem. That is,
> $$z:=\max_{\lVert b\rVert_1\leq1, \lVert b\rVert_1\leq1, b\succeq 0, <k,1>=0}b^\top A k = \max_{\lVert b\rVert_1\leq1, b\succeq 0}\max_{ \lVert k\rVert_1\leq1, <k,1>=0}b^\top A k= \max_{\lVert b\rVert_1\leq1, b\succeq 0}\lVert b^\top A \rVert_{sp},$$
> where $\lVert x \rVert_{sp} = \frac{\max_{x_i}-\min_{i}x_i}{2}$ is span-norm. Now, we have to maximize the above span-norm over the simplex ($\lVert b\rVert_1\leq1, b\succeq 0$), and we can easily show that
> $$z = \max_{i}\lVert A(i,\cdot)\rVert_{sp}.$$
> We will add this discussion in detail in the final version. To summarize, non-rectangular robust MDP with $L_1$ bounded ball can be efficiently be solved in polynomial time.
>
> > For $p=\infty$, non-rectangular uncertainty set degenerates to sa-rectangular one, which is already efficiently solvable [19].
>
> > For $p=2$, we proposed a heuristic spectral Algorithm 2, to solve the above sub-problem. And  Figure 6, Figure 7 and Table 3 shows the superiority of its performance over numerical method such as scipy. Further, we showed in the previous response that this method also yields much better performance  than random search even when uncertainty radius $\beta$ is big-enough.
>
> > For other $p\neq 1,2$, we are not sure if the above sub-problem is efficiently sovable or not. As a similar problem without positivity and mean-zero constraints,
> $$ A_{p\to q} := \max_{\lVert b\rVert_p\leq1, \lVert k\rVert_p\leq1}b^\top A k = \max_{\lVert b\rVert_p\leq1}\lVert b^\top A \rVert_q,$$
> is very well studied known as Matrix p-norms. And there are  several negative and constant approximation results are known in the literature  for  Matrix p-norm problem for $p\neq 1,2,\infty$ [Hendrickx and Olshevsky].
>
> To summarize, for $L_1$ bounded uncertainty set are efficient and solutions can be obtained in the close form. And for $p=\infty$, it is equvialent to sa-rectangular case which is easy. For $p=2$, we provided heuristic algorithm that works well in practice and also generalizes well beyond theoretically applicable uncertainty radius $\beta$. However, we believe that for $p=2$, it is possible to provably solve this sub-problem via standard methods as second-order cone optimization or quadratic programming, but we are not aware of the exact positive nor negative result.  And for other $p$, we are not sure if the efficient solution exists.
>
> We thank the reviwer again for this thoughful question. We will gladly add this discussion in the final version.
>
> [Hendrickx and Olshevsky] @article{DBLP:journals/corr/abs-0908-1397, author  = {Julien M. Hendrickx and Alexander Olshevsky},title        = {Matrix P-norms are NP-hard to approximate if p {\textbackslash}neq1,2{\textbackslash}infty},journal      = {CoRR},volume       = {abs/0908.1397},year         = {2009}}

---

> > ### Comment · Reviewer_oaPp · 2025-08-07
> >
> > Thank you for your thoughtful response. I have no more questions and am happy to increase my score to reflect this.

---

### Note · Authors · 2025-08-12

We thank all reviewers, the AC, and the SAC for their time.

We are pleased that the reviewers appreciated our polytime dual approach to addressing non-rectangular robust MDPs, a problem known to be NP-Hard in general. We clarified the main concerns as follows:

**A) Performance under large uncertainty radius (\$\beta\$):** While the accuracy may degrade slightly for large \$\beta\$, the computational complexity of our method remains unchanged. As shown in our response to Reviewer OaPp, even with large \$\beta\$, our approach identifies a significantly better worst-case kernel within 2–4 seconds, compared to over 400 minutes required by random sampling.

**B) Complexity and performance across different \$p\$ values:** Our Algorithm 1 converges in \$O(\log \epsilon^{-1})\$ iterations for all \$p\$, where \$\epsilon\$ is the desired accuracy. The core subroutine is efficiently solvable: in closed form for \$p=1\$ (see Reviewer OaPp response), via a spectral heuristic for \$p=2\$, and efficiently for \$p=\infty\$ as it aligns with the SA-rectangular case. For other $p$, we believe it may be NP-hard.

In summary, we present the first efficient method for policy evaluation in non-rectangular robust MDPs, opening up promising directions for future research in a previously intractable setting.

---

### Decision · Program_Chairs · 2025-09-17

**Decision:**

Accept (poster)

**Comment:**

This paper studies robust policy evaluation under non-rectangular uncertainty sets, which is generally NP-hard, and introduces a special class of \ell_p-bounded sets that avoid these complexity barriers. By decomposing them into infinitely many sa-rectangular sets, the authors derive a novel dual formulation and an efficient algorithm for robust policy evaluation in these settings.

I believe that the paper makes a strong theoretical contribution by providing the first polynomial-time algorithm for a certain coupled transition uncertainty sets. The dual reformulation and efficient algorithm are elegant, broadly applicable, and supported by proofs and significant empirical speed-ups. While the results have some limitation (e.g., requiring a small uncertainty radius), I agree with the authors that they serve as an important step in the right direction and encourage further exploration in this area. Reviewer p4ao (who gave the lowest rating 3) expressed concerned with the motivations of the proposed uncertainty set, the novelty of the analysis, as well as the characterization of the convergence rate of the algorithm, but the authors provided convincing rebuttal to these points (especially the motivations and the convergence rate). Therefore, even if Reviewer p4ao did not follow up on the discussion nor change their score, I still recommend accept.